



**Isoprene and monoterpene emissions from alder, aspen and spruce short**
**rotation forest plantations in the UK**
Gemma Purser[*,1,2], Julia Drewer[1], Mathew R. Heal[2], Robert A. S. Sircus[2], Lara K.
Dunn[2], James I. L. Morison[3]
[1] UK Centre for Ecology & Hydrology, Bush Estate, Penicuik, Midlothian, EH26 0QB, UK
[2] School of Chemistry, University of Edinburgh, Joseph Black Building, David Brewster Road,
Edinburgh, EH9 3FJ, UK
[3] Forest Research, Alice Holt Lodge, Farnham, Surrey, GU10 4TT, UK
*Corresponding author

## Abstract

An expansion of bioenergy has been proposed to help reduce fossil-fuel greenhouse gas emissions,
and short-rotation forestry (SRF) can contribute to that expansion. However, SRF plantations could
also be sources of biogenic volatile organic compound (BVOC) emissions, which can impact on
atmospheric air quality. In this study, emissions of isoprene and 11 monoterpenes from the branches
and forest floor of hybrid aspen, Italian alder and Sitka spruce stands in an SRF field trial in central
Scotland were measured during two years (2018–2019) and used to derive emission potentials for
different seasons. Sitka spruce was included as a comparison as it is the most extensive plantation
species in the UK. Winter and spring emissions of isoprene and monoterpenes were small compared
to those in summer. Sitka spruce had a standardised average emission rate of 15 µg C g$^{-1}$ h$^{-1}$ for
isoprene in the dry and warm summer of 2018, more than double the emissions in 2019. However,
standardised average isoprene emissions from hybrid aspen were similar across both years,
approximately 23 µg C g$^{-1}$ h$^{-1}$ and standardised average isoprene emissions from Italian alder were
very low. Average standardised total monoterpene emissions for these species followed a similar
pattern of higher emissions in the warmer year: Sitka spruce emitting 4.5 µg C g$^{-1}$ h$^{-1}$ and 2.3 µg C g$^{-1}$





h$^{-1}$ for 2018 and 2019, aspen emitting 0.3 µg C g$^{-1}$ h$^{-1}$ and 0.09 µg C g$^{-1}$ h$^{-1}$ and Italian alder emitting,
1.5 µg C g$^{-1}$ h$^{-1}$ and 0.2 µg C g$^{-1}$ h$^{-1}$, respectively. In contrast to these foliage emissions, the forest
floor was only a small source of monoterpenes, typically one or two orders of magnitude lower than
foliage emissions on a unit ground area basis.  Estimates of total annual emissions from each
plantation type per hectare were derived using the MEGAN 2.1 model. The modelled total BVOC
(isoprene and monoterpenes) emissions of SRF hybrid aspen plantations were approximately half
those of Sitka spruce for plantations of the same age. Italian alder SRF emissions were 20 times
smaller than from Sitka spruce. The expansion of bioenergy plantations to 0.7 Mha has been
suggested for the UK to help achieve "net-zero" greenhouse gas emissions by 2050. The model
estimates show that with such an expansion total UK BVOC emissions would increase between  <1%
and 35%, depending on the tree species planted. Where increases might be small on a national
scale, regional increases might have a larger impact on local air quality.

## 1. Introduction

The UK has committed to reducing its carbon dioxide ($CO_2$) emissions to meet net-zero greenhouse
gas emissions targets by 2050, and increasing bioenergy use is seen as a substantial pathway to this.
Bioenergy was the largest contributor to renewable energy within the UK in 2018, accounting for 7%
of the primary energy supply (Renewable Energy Association, 2019) and it has been suggested that
this could grow to 15% by 2050 (Committee on Climate Change, 2019). Solid biomass, in the form of
wood pellets, chips, and agricultural and forestry residues, is the primary type of biomass used to
generate heat and electricity, accounting for 60% of bioenergy in 2016 (IEA Bioenergy, 2018).
However, the majority of the 7.2 million tonnes of wood pellets burned in the UK in 2018 came from
imports from North America (Renewable Energy Association, 2019). A larger contribution from
domestic supply of bioenergy in the UK is required.




Currently the most common bioenergy crops in the UK are coppiced willow and *Miscanthus*, a
perennial grass. Only 1.6% of arable land has been used in recent years for biomass in the UK
(DEFRA, 2019) but this needs to increase (Committee on Climate Change, 2019). Short rotation
coppice (SRC), in which woody plants such as willow is grown on a 3–4 year cycle, provides high-
volume short-term biomass yields but typically produces biomass of lower calorific value compared
to short rotation forest (SRF). In SRF, single stemmed trees are grown over 10–20 years for either
biomass or timber. This produces a better timber to bark ratio for higher biomass yields, is easily
harvested and offers increased flexibility to growers in times of uncertain biomass markets (Keith et
al., 2015; Leslie et al., 2012; McKay, 2011). The recent Committee on Climate Change report (2020)
suggested that 0.7 million hectares of energy crops (*Miscanthus*, SRC or SRF) should be grown in the
UK by 2050 as a 'Further Ambition' scenario in order to achieve net zero emissions and increase the
domestic supply of biomass.

In 2010, Forest Research established SRF trials across the UK to determine biomass yields and assess
the environmental impact of SRF (Harrison, 2010). The trials included a number of broadleaf tree
species (hybrid aspen, red alder, common alder, Italian alder, sycamore, horse chestnut, eucalyptus
spp.) and the two conifer species Sitka spruce and hybrid larch (Harrison, 2010). Sitka spruce is the
most widely grown conifer tree species in the UK and a key plantation species. SRF plantations have
previously been assessed for their environmental impact in the UK and Ireland (Keith et al., 2015;
McKay, 2011; Tobin et al., 2016), but not for their potential future impacts on air quality in the UK,
which is the focus of this work.



Trees are known sinks for $CO_2$ but can also be sources of other trace gases such as volatile organic
compounds (VOCs) (Monson and Fall, 1989; Went, 1960). VOCs are emitted by tree foliage as a
means of communication, plant defence against herbivory and during environmental stress such as
heat or drought. Other sources of VOCs within a forest may include wood, litter, soils, fruits, flowers
and roots (Dudareva et al., 2006). Emitted VOCs include, in particular, isoprene and monoterpenes,
and their aliphatic, aromatic and oxygenated derivatives. These compounds are highly reactive in the
atmosphere and contribute to the formation of tropospheric ozone in the presence of nitric oxide
(NO) (Atkinson and Arey, 2003). Terpene composition has been found to be an important factor in
the magnitude of ozone production (Bonn et al., 2017). Ground-level ozone is a concern for
agriculture and natural ecosystems as it causes leaf damage, reduced plant growth (Emberson, 2020;
Fares et al., 2013; Felzer et al., 2007) and is also a pollutant with impacts on human-health and as a
greenhouse gas (UNEP/WMO, 2011). In addition, intermediates of VOC oxidation may act as
condensation nuclei for the formation of secondary organic particles (Carlton et al., 2009), another
atmospheric pollutant with detrimental effects on human health (Fuzzi et al., 2015).

The emissions of VOCs from plants are dependent upon a range of factors (which vary with emitting
source and type of VOC) including species, plant age and environmental conditions such as light and
temperature (Guenther et al., 1991; Monson and Fall, 1989) and, in the case of forest floor
emissions, soil moisture, ambient temperature, soil type and the activity of the soil microbiome
(Peñuelas et al., 2014). If the area of bioenergy crops expands, determining their VOC emissions
becomes necessary for the wider assessment of air quality for a given region. Willow, a current UK
bioenergy crop grown as SRC is a known emitter of VOCs (Morrison et al., 2016), but there is a lack
of literature data generally for VOC emissions from trees in SRF plantations and from the forest
floor.





100 In this study we focus on determining the contribution of the BVOC emissions from the two species

101 with the largest growth during SRF trials in the UK: hybrid aspen and Italian alder (McEvoy, 2016;

102 McKay, 2011; Parratt, 2018). In addition, we measured the BVOC emissions for young Sitka spruce

103 plantations, also grown at the same location, as a comparison. Measurements were made in a

104 plantation species-trial in central Scotland. Using dynamic enclosure sampling of BVOCs onto

105 absorbent cartridges, the contribution of both foliage and forest floor emissions were measured

106 simultaneously on occasions to form a plantation-scale assessment of BVOC emissions. The data

107 were then used with the MEGAN 2.1 model (Guenther et al., 2012) to derive an estimate of the

108 potential total annual contribution of expanded SRF to UK BVOC emissions.

109

## 110 2. Methods

### 111 2.1 Field site description

#### 112 2.1.1 Tree species and planting

113 Measurements were made at East Grange, Fife, Scotland (Lat/Lon (WGS84) 56° 05' 21" N,

114 003° 37' 52" W), elevation 45–60 m, one of the 16 SRF trial locations established by Forest Research

115 (Harrison, 2010; Stokes, 2015). Soil type and texture at the site is surface-water gley and sandy silty

116 loam respectively, containing 4.9% clay, 53.0% silt and 42% sand (Drewer et al., 2017; Keith et al.,

117 2015). In 2010, the ex-agricultural site was planted with a single block of 40 randomised tree species

118 plots and 8 control plots. Each plot (20 m x 20 m) consisted of a single species containing 200 trees

119 with a 2 m x 1 m spacing arrangement (Harrison, 2010). Ten species were planted, and the two

120 broadleaved species with the best survival and growth rates across the trials in the first six years,

121 hybrid aspen (*Populus tremula L. x tremuloides* Michx.) and Italian alder (*Alnus cordata* Desf.), were

122 selected for the measurements here, along with Sitka spruce (*Picea sitchensis* Bong. Carr, produced

123 by vegetative propagation) (McEvoy, 2016; Parratt, 2018). After initial establishment of the young



saplings, the site remained unmanaged. Branch and forest floor sampling chambers were installed in
single south facing plots of each species.

### 2.1.2    Meteorological data

Meteorological data were collected from an unplanted plot in the middle of the site between May
2018 and July 2019. Minimum and maximum soil temperature (T107, Campbell Scientific, Shepshed,
Leics, UK), air temperature and relative humidity (HMP45C, Campbell Scientific) were monitored
hourly. In addition, photosynthetic active radiation (PAR, SKP 215 Quantum Sensor, Skye
instruments, Llandrindod Wells, UK) was measured at the same site every 5 minutes. Monthly
averages and ranges are provided in Supplementary Information S1. Occasional power failure at the
site led to some missing data. For the modelling of BVOC emissions using Pocket MEGAN 2.1 excel
beta 3 calculator (Guenther 2012) the missing PAR and average temperature data were replaced by
measurements from the Easter Bush site of the UK Centre for Ecology & Hydrology lying 45 km to
the south east (Lat/Lon (WGS84) 55° 51' 44" N, 003° 12' 20" W). A summary of the combined East
Grange and Easter Bush data used in the model can be found in Supplementary Information S2.

The climate in east Scotland is colder, with fewer sunshine hours than in the south of England. To
encompass these climate differences, meteorological data from Alice Holt forest (51°09'13"N ,
000°51'30"W), Hampshire, in southern England recorded during 2018 and 2019 was also used for
the modelling and scaling up of the measured BVOC emission potentials from this study. A summary
of the PAR and air temperature data for this field site is given in Supplementary Information S3.



### 2.2 Sampling enclosures


Branch sampling was conducted on the spruce, aspen and alder plantation plots on a total of 16, 11
and 13 days respectively between March 2018 and July 2019. The plantation floor sampling was
conducted on a total of 18 (spruce and alder) and 20 days (aspen) for the same plots during the
same period.

#### 2.2.1 Forest floor enclosures


Forest floor in this context includes soils, leaf litter, fallen small twigs/branches and flowers,
understorey vegetation, microorganisms and underground biomass that may all be sources of BVOC
from the ground of the plantation. A static chamber method was used for the plantation floor
enclosures. Polyvinylchloride plastic soil collars (with a flange), 40 cm diameter x 18 cm high, were
installed per tree species plot prior to sampling (Asensio et al., 2007c, 2007b; Greenberg et al., 2012;
Janson, 1993) and remained in the ground for the duration of the experiment. One or two collars
were installed in 2017 and used during 2018. Additional collars were installed during 2018 resulting
in a total of three soil collars per plot for the 2019 sampling. The collars were placed towards the
centre of each plot to reduce the likelihood of plant debris from other plots contaminating them.
Leaf litter and understorey vegetation were not removed from the collars prior to sampling to reflect
actual changes in BVOC emissions with changes in the forest floor composition through the seasons.
A clear acrylic lid (with a foam lined flange), 40 cm diameter x 22.5 cm high, was placed over the soil
collar during sampling periods only, enclosing a total chamber volume of 51 L. The lid was sealed
using clamps around the rim. A small 12 V axial fan (RS components Ltd, Colby, UK),  4 cm x 4 cm x 1
cm, was attached to the chamber lid to mix the air inside the chamber (Janson, 1993). Samples of
BVOC in the enclosed air were collected through PTFE tubing onto a 6 mm OD stainless steel
automated thermal desorption (ATD) cartridge (PerkinElmer, Waltham, MA, USA) packed with 200
mg Tenax TA 60/80 (11982 SUPELCO, Sigma-Aldrich, St Louis, MO, USA) and 100 mg Carbotrap 20/40





(20273 SUPELCO, Sigma-Aldrich) at a flow rate of 0.2 L min$^{-1}$ using a handheld pump (210-1003MTX,
SKC ltd, Blandford Forum, UK). Samples were collected for 30 min after closure, equating to a total
sample volume of 6 L. Pressure compensation was maintained through a small hole in the side of the
chamber to prevent negative pressure inside the chamber and potential degassing of air from the
soil pores. Ambient air samples were collected concurrently with the chamber sample in order to
quantify BVOC emissions from the forest floor by difference. This is discussed further in Section
2.5.2. No ozone filter was used during sampling so amounts of some monoterpenes may have been
reduced by reaction with ozone (Ortega et al., 2008). However, it has also  been suggested that
ozone may be lost by dry deposition onto the chamber walls in the first minute (Janson et al., 1999).
Chamber air temperature (Electronic Temperature Instruments Ltd, Worthing, UK) and humidity
(Fisherbrand™ Traceable™ Humidity Meter, Fisher Scientific, Loughborough, UK) were measured at
the end of the 30 min sample collection period.
Volumetric soil moisture (ML3 ThetaProbe Soil Moisture, Delta T, Cambridge, UK) was measured at
three locations around each chamber and soil temperature was measured at a single location at 7
cm depth close to, but outside the soil collar to avoid disturbance of the forest floor. Both
measurements were performed after sample collection to prevent perturbation of the ambient air
sample.

2.2.2    Branch enclosure
A dynamic chamber method was used for branch enclosures. Three sample points were established
per tree species plot and used to mount a removable flow-through acrylic chamber (Potosnak et al.,
2013), 53 L in volume. The chambers were set up during each sampling visit and used to enclose a
single branch, alternating between three similar branches per tree species. Ambient air flow was
delivered from an oil-free double-ended diaphragm pump (Capex V2, Charles Austen pumps Ltd,
Surrey, UK) through PTFE tubing (Morrison et al., 2016; Purser et al., 2020) at a flow rate of 10 L min$^{-}$





[1] to obtain the desirable air exchange rate of 4-5 minutes (Ortega and Helmig, 2008). In addition, the
chamber contained a small 12 V axial fan (RS components Ltd, Colby, UK), 8 cm x 8 cm x 2.5 cm, to
ensure mixing of air inside the chamber.

After set-up, the branch enclosure was left for a period of 30 min to attain a steady state. Both
inside and outside of the enclosure were then sampled concurrently for 30 min at a flow rate of 0.2 L
min$^{-1}$ (total sample volume of 6 L) using a handheld pump (210-1003MTX, SKC Ltd, Blandford Forum,
UK). In cases of low light levels, low temperatures or smaller volumes of foliage, the sampling time
was sometimes extended (up to 60 minutes) to ensure sufficient sample was collected on the
sample cartridge. Multiple sequential samples were taken over a given day. All enclosure sample
tubes were stored in a fridge at 4 °C until analysis.

After BVOC sample collection, the leaves inside the chamber were counted and a representative
subsample of approximately 10% of the total number of leaves on the measured branch removed
from a nearby branch. The leaves were dried at 70 °C until constant mass, typically after 48 h. In the
case of the Sitka spruce subsidiary branches were used. Measurements of chamber temperature and
relative humidity (CS215, Campbell Scientific, Shepshed, UK) were made each minute during
sampling. In addition, PAR (SKP 215 PAR Quantum Sensor, Skye instruments, Llandrindod Wells, UK)
was measured outside but next to the branch chamber with measurements made every minute. The
chambers had 85% transparency to PAR (400–700 nm), so the measured PAR values were
correspondingly adjusted to represent the illumination conditions inside the chamber.





## 2.3 BVOC analysis

The BVOC samples collected on the sorbent were analysed using gas chromatography-mass
spectrometry (GC-MS) with a two-stage automatic thermal desorption unit (ATD 400, Perkin-Elmer,
Wellesley, MA, USA) using the method described in Purser et al. (2020). Calibration was carried out
using standards (from Sigma-Aldrich, Gillingham, UK) of the monoterpenes α-pinene, β-pinene, d-
limonene, α-phellandrene, β-phellandrene, 3-carene,  camphene, γ-terpinene and β-myrcene, and
the monoterpenoids (monoterpene-based compounds with, for example, additional oxygen or
missing a methyl group) eucalyptol and linalool prepared as a mixed stock solution of 3 ng μL$^{-1}$ in
methanol. Aliquots of 1, 2, 3 and 4 μL of the mixed monoterpene stock solution were pipetted
directly onto sample tubes under a flow of helium to produce a range of mixed monoterpene
standards of 3, 6, 9 and 12 ng. Isoprene standards were prepared by direct sampling onto a sorbent
tube from a certified 700 ppbv gas standard (BOC, UK) for 10, 30, 45 and 60 s using a sample pump
(210-1003MTX, SKC ltd, Blandford Forum, UK) producing standards of 65, 198, 296 and 395 ng. Note
that mass loadings of isoprene and monoterpene calibration standards were calculated to greater
precision than quoted above but are shown here as nominal values for ease of discussion.

Unknown peaks in sample chromatograms were identified by comparison to the internal library of
the GC-MS (National Institute of Standards and Technology) and by comparison with the retention
time of the standard. The limit of detection (LOD) of the calculated emissions ranged from 0.12-0.35
μg C g$_{dw}$$^{-1}$ h$^{-1}$ for the branch chambers and 0.47-1.4 μg C m$^{-2}$ h$^{-1}$ for the forest floor chambers.
Uncertainties on an individual calculated emission rates were 16% for isoprene and 17% for
monoterpenes, which were derived via error propagation methods described in Purser et al. (Purser
et al., 2020).



## 2.4 Calculation of standardised emissions

### 2.4.1    Forest floor BVOC emissions

As no substantial isoprene emissions were observed during an initial assessment, only

monoterpenes were quantified from the forest floor. Monoterpene emissions from the forest floor

($F_{floor}$) were calculated as µg carbon for a given compound per ground surface area  (µg C m$^{-2}$ h$^{-1}$)

using Eq. (1), where $C_{sample}$ is the concentration of a monoterpene inside the chamber (µg C L$^{-1}$),

$C_{ambient}$ is the concentration of a monoterpene in the ambient air outside the chamber (µg C L$^{-1}$), $A$ is

the area of forest floor inside the chamber (m$^2$), $V$ is the volume inside the chamber, and , $t$ is the

sampling duration (mins).

$$F_{floor} = \frac{[C_{sample} - C_{ambient}] \times V \times 60}{A \times t} \tag{1}$$

In some cases, the concentration in ambient air was larger than inside resulting in a negative

emission value, i.e. a net uptake.

### 2.4.2    Branch scale BVOC emissions

The isoprene or monoterpene emission ($F_{branch}$) from an enclosed branch was calculated as µg

carbon (C) for a given compound per leaf dry mass basis, µg C g(dw)$^{-1}$ h$^{-1}$, using Eq. (2),  where $f$ is the

flow rate through the chamber (L min$^{-1}$) and $m$ is the dry mass (g) of foliage inside the chamber.

$$F_{branch} = \frac{[C_{sample} - C_{ambient}] \times f}{m} \tag{2}$$

Isoprene emissions have previously been shown to be controlled by both light and temperature and

can be standardised to 30 °C and 1000 µmol m$^{-2}$ s$^{-1}$, respectively (Guenther et al., 1993). Average

chamber air temperature and PAR for each period of sample collection were therefore used to

standardise the measured $F_{branch}$ emissions for isoprene (Eq. (3), (4) and (5)) and monoterpenes (Eq.



6) to facilitate comparison between this study and previous literature. The algorithms developed in
Guenther et al. (1993) are subsequently referred to as G93.
The standardised isoprene emission rate $F_{\text{isoprene}}$ at 30 °C and 1000 µmol m$^{-2}$ s$^{-1}$ PAR is a function of
the measured emission $F_{\text{branch}}$, a term $C_L$ to correct for the effect of light and a term $C_T$ to correct for
the effect of temperature Eq. (3).
$$F_{\text{isoprene}} = \frac{F_{branch}}{C_L \times C_T} \qquad (3)$$
The light-correction term $C_L$ is calculated from Eq. (4) where $\alpha$ = 0.0027 and $C_{L1}$ = 1.066 are empirical
coefficients in G93 and $L$ is the experimentally-measured average PAR (µmol m$^{-2}$ s$^{-1}$) during sampling.
$$C_L = \frac{\alpha C_{L1}\, L}{\sqrt{1 + \alpha^2\, L^2}} \qquad (4)$$
The temperature-correction term $C_T$ is calculated using Eq. (5) in which the terms $C_{T1}$ (95000 J mol$^{-1}$),
$C_{T2}$ (230000 J mol$^{-1}$) and $T_M$ (314 K) are all empirically-derived coefficients from G93. $R$ is the molar
gas constant 8.314 J K$^{-1}$ mol$^{-1}$, $T$ is the average air temperature (K) during sampling, and $T_s$ is the
standardised temperature of 303.15 K, equivalent to 30 °C.

$$C_T = \frac{exp\frac{C_{T1}(T - T_S)}{RT_S T}}{1 + exp\frac{C_{T2}(T - T_M)}{RT_S T}} \qquad (5)$$

Monoterpene emissions from branch chambers, $F_{\text{branch}}$ were standardised to temperature based on
the calculations from Guenther et al.  (1993) using Eq. (6). $T_s$ is the standard temperature (303 K) and
$T$ is the average air temperature during sampling. $F_{\text{monoterpene}}$ is the standardised monoterpene
emission rate (µg C g$_{\text{(dw)}}^{-1}$ h$^{-1}$) and $F_{\text{branch}}$ is the measured monoterpene emission rate (µg C g$_{\text{(dw)}}^{-1}$ h$^{-1}$).



$$F_{\text{branch}} = F_{\text{monoterpene}} \exp(\beta(T - T_s)) \qquad\qquad (6)$$

Standardised isoprene and monoterpene emission rates from sequential samples calculated for a given day were then averaged to give a single standardised branch emission rate per tree species per measurement day. In addition, daily measurements were grouped into seasons to give a standardised emission potential per season, $F_{\text{b\_season}}$.

## 2.5 LAI determination

A Leaf Area Index (LAI) meter (LAI-2000 plant canopy analyser, LI-COR, Inc., Lincoln, NE, USA) was used to provide data to estimate a density of foliage, $m^2_{\text{leaf}} \, m^{-2}_{\text{ground}}$, for each species during two separate days, two weeks apart in July 2018, assumed to be the time of maximum foliage density (Ogunbadewa, 2012). LAI determinations were made in three hybrid aspen, two Sitka spruce and one Italian alder plots. Two above-canopy and eight below-canopy points were measured per plot, with a mixture of within and between row measurements. Where more than one plot was measured for a species, the average LAI is reported.

## 2.6 Scaling up from emission per mass of foliage to an emission per area of ground of plantation

The standardised emissions of isoprene and monoterpenes from the canopy ($\mu g \, C \, m^{-2}_{\text{ground}} \, h^{-1}$), $F_{\text{foliage}}$, was determined using Eq. (7), multiplying standardised summertime branch emission measurements ($F_{\text{b\_summer}}$) calculated in Section 2.5.2 with literature values of the leaf mass per leaf area (LMA) for each tree species (Table 1) and the measured LAI. As there was limited LMA data for Italian alder under climate conditions relevant for the UK, additional values were taken from literature on common alder (*Alnus glutinosa*). The LMA multiplied by the LAI gives the mass of



foliage per unit area of ground, known as the foliar biomass density. The calculated foliar biomass
density values in Table 1 for hybrid aspen (329 g m$^{-2}$) and Italian alder (315 g m$^{-2}$) are very similar to
the 320 g m$^{-2}$ (Karl et al., 2009) and 375 g m$^{-2}$ (Geron et al., 2000) used in previous modelling studies
for these two tree species. For Sitka spruce the foliage biomass density used here (619 g m$^{-2}$) is
about half that for the same species in previous modelling studies, 1500 g m$^{-2}$ (Geron et al., 2000;
Karl et al., 2009) and reflects the immature Sitka spruce stand not yet achieving a closed canopy.

$$F_{\text{foliage}} = F_{\text{b\_summer}} \times LMA \times LAI \qquad\qquad (7)$$

For times when the plantation canopy consisted of flowers only (catkins) or early leaf emergence,
during the months February to April on deciduous species, a different approach had to be applied. In
these instances the LAI was either reduced to reflect the canopy during leaf emergence or the
following estimate for catkins was applied. We assumed that there were approximately 66 catkins
per m$^{-2}$ per ground area of the plantation canopy based on similar catkin forming species
(Boulanger-Lapointe et al., 2016). This equates to a catkin biomass density, for converting from
branch-scale to canopy-scale purposes, of 8.98 g m$^{-2}_{\text{ground}}$ based on the average mass of an alder
catkin measured during our study.

Branch measurements made during April when leaves were young were assigned lower LAI values,
such as 1.06 for Hybrid aspen and 0.81 for Italian alder. This modification of LAI through the year
(Table 2) was based on multiple LAI measurements taken across the year in a deciduous forest stand
in the UK (Ogunbadewa, 2012) in which by late-April (day of year 120) a quarter of the maximum LAI
was reached and half the maximum LAI by mid-May (day of year 141). In that study the maximum
LAI was recorded in mid-July (day of year 210).



**Table 1 – Leaf mass per area data for calculating foliage emission rates per plantation ground area.**

| Tree species | LMA / g m$^{-2}_{leaf}$ | Literature source | Country of origin of literature measurement | Forest type | Stand age / years | Measured LAI during this study | Foliar biomass density / g m$^{-2}_{ground}$ |
|---|---|---|---|---|---|---|---|
| Hybrid aspen | 98.0 | (Tullus et al., 2012) | Estonia | Trial plantation | 4 | | |
| | 73.5 | (Yu, 2001) | Finland | Clone trial | 1.5 | | |
| | 61.7 | (Johansson, 2013) | Sweden | SRF Plantation | 15-23 | | |
| Average RSD / % | 77.7 24 | - | - | - | - | 4.24 | 329 |
| Sitka spruce | 222 | (Norman and Jarvis, 1974) | NS | Plantation | NS | | |
| | 160 | (Meir et al., 2002) | Scotland | Plantation | 13 | | |
| | 200 | (Foreman, 2019) | Ireland | Greenhouse trial | 3 | | |
| Average RSD / % | 194 16 | - | - | - | - | 3.19 | 619 |
| Italian alder | 114** | (Leslie et al., 2017) | England | Trial Plantation | 2 | | |
| | 102* | (Foreman, 2019) | Ireland | Greenhouse trial | 2 | | |
| | 75.1** | (Johansson, 1999) | Sweden | Plantation | 21-91 | | |
| Average RSD / % | 97.0 21 | - | - | - | - | 3.25 | 315 |

*Average of sun and shade leaves. NS = Not specified, RSD = relative standard deviation.
**Measurements from common alder (*Alnus glutinosa*)


## 2.7  From canopy emission to total annual emissions per hectare and the influence of
## increasing biomass planting on total UK BVOC emissions
Standardised foliage emission rates, $F_{foliage}$, for summer 2018 and 2019 (Table 3) were input to the
Pocket MEGAN 2.1 excel beta 3 calculator (Guenther et al., 2012) with hourly average PAR and
temperature data from East Grange (gap filled with UKCEH site data), LAI and the other variables
given in Table 2. For a detailed description of the equations and algorithms used in MEGAN 2.1 see
Guenther et al. (Guenther et al., 2006, 2012). The model adjusts the standardised emission rate
input in accordance with air temperature and PAR from the meteorology inputs per hour to produce
a likely emission rate for the plantation. Input LAI measurements for alder and aspen were scaled in
spring and autumn by 25% and 50% to simulate leaf emergence and senescence (Table 2). The LAI of
Sitka spruce was assumed to remain constant through the seasons although it is recognised there
will be a small increase in the spring, and a later decline. No LAI measurements were made in 2019





therefore 2018 measurements were used. The function that accounts for the effect of both the
previous 24 hours and 240 hours of light on the calculated emissions was applied in the model.  The
latitude was set to 56⁰ for Scotland and 51⁰ for England and the vegetation cover was set to 1. The
functions in MEGAN2.1 that allow for consideration of soil moisture and $CO_2$ concentrations were
not used due to a lack of continuous data available for the field sites. The monoterpenes in the
model were calculated using the single value for average total monoterpene from East Grange and
using the category named "other monoterpenes". An assumption was made that the emissions were
driven by temperature only and no light specific emission fraction was specified due to the different
behaviours of the collective "total monoterpenes". Any other model input parameters remained as
default.

The model output of hourly isoprene and total monoterpene emissions were summed to give annual
emissions per $m^2$ of SRF plantation. The combined average total annual emission rate encompassing
both years of emission potentials (2018 and 2019) and meteorology from two contrasting UK sites
(E. Scotland and S.E. England), for each SRF species, was then compared to literature values for the
estimated annual UK isoprene and monoterpene emissions and combined total BVOC emissions.










**Table 2 – Seasonal time course of leaf area index (LAI) for estimating annual VOC emissions for**
**different species plots at East Grange, Fife, Scotland, using MEGAN 2.1 model.**

| Date | Day of year | Sitka LAI | Aspen LAI | Alder LAI |
|---|---|---|---|---|
| 1st January | 1 | 3.19 | 0 | 0 |
| 19th February | 50 | 3.19 | 0 | 0 |
| 31st March | 90 | 3.19 | 0 | 0 |
| 19th April | 109 | 3.19 | 1.06 | 0.81 |
| 30th April | 120 | 3.19 | 2.12 | 1.63 |
| 1st June | 152 | 3.19 | 3.18 | 2.43 |
| 15th July | 196 | 3.19 | 4.24 | 3.25 |
| 1st August | 213 | 3.19 | 4.24 | 3.25 |
| 1st September | 244 | 3.19 | 3.18 | 2.43 |
| 20th October | 304 | 3.19 | 1.06 | 0.81 |
| 31st October | 334 | 3.19 | 0 | 0 |
| 31st December | 366 | 3.19 | 0 | 0 |



**Table 3 – Input parameters for estimating annual BVOC emissions for different SRF species plots at**
**East Grange, Fife, Scotland using the MEGAN 2.1 model.**

| | Spruce | | Aspen | | Alder | |
|---|---|---|---|---|---|---|
| Emission rate (per unit ground area) | 2018 | 2019 | 2018 | 2019 | 2018 | 2019 |
| Isoprene / mg $m^{-2}_{ground}$ $h^{-1}$ | 9.31 | 4.23 | 7.74 | 7.30 | 0.01 | 0.01 |
| Total monoterpene / mg $m^{-2}_{ground}$ $h^{-1}$ | 2.81 | 1.45 | 0.09 | 0.03 | 0.22 | 0.07 |



## 382  3. Results and discussion


### 384  3.1 Field observations of seasonality

The measured BVOC emissions were assigned to seasons as follows: winter (21[st] December – 19[th]
March), spring (20[th] March – 07[th] June), summer (08[th] June – 22[nd] September) and autumn (23[rd]
September – 20[th] December). 2018 is classified here as a dry year, being 25% drier at the East
Grange field site than the 30 year average for the area (Met Office, 2020). In contrast, 2019 was 50%
wetter than the 30 year UK average. In 2019, catkins were fully developed on the hybrid aspen and
Italian alder branches by February, but bud burst and leaf emergence was not observed until mid-





April (19th). This was two weeks later than in 2018. The first new growth on the Sitka spruce was
observed at the end of April (29th). Based on these differences in phenology at the site,
measurements taken on 7th June 2019 was still categorised as spring.

For the forest floor it was noted that the soil temperatures during summer 2018 were higher than in
2019. After several dry weeks in spring and summer in 2018, the first significant rainfall event since
May was noted as 14th July, and some leaf fall in the Italian alder and hybrid aspen plots was
observed by the end of July. By February 2019, no leaf litter from the previous autumn season was
observed on the forest floor of the plots except for those of Sitka spruce. Rapid understorey growth
identified as hogweed (*Heracleum sp)* quickly developed from late April (29th) and by early June (7th)
completely covered the forest floor in the alder plots. The hybrid aspen and Sitka spruce plots during
both 2018 and 2019 had minimal understorey vegetation by comparison.
## 3.2 Leaf area index
The LAI of 3.19 for our 8-y old Sitka spruce plantation (Table 1) is lower than the value of 4.33
predicted for a 10-y old plantation from allometric relationships (Tobin et al., 2007). However, our
measured LAI reflects a canopy not yet fully closed and the differences in site conditions are likely to
produce different growth rates.
A maximum LAI of 4 was reported for a 9-y old aspen (*Populus tremuloides Michx.*) plantation in
Canada (Pinno et al., 2001), which compares well with the LAI of 4.24 measured here (Table 1).
A 4-y old SRF plantation of Italian alder established in Ireland that was also measured in July gave an
LAI of 2.8 or 3.4 for a 2 x 2 m or a 1 x 1m plant spacing respectively (Foreman, 2019). Other alder
species such as common (or black) alder (*Alnus glutinosa*) and grey alder (*Alnus incana*) in Sweden
had LAI values of 2.85 and 3.04, respectively; all comparable to the Italian alder LAI of 3.25 measured
here (Table 1). A study of SRF planting density trials in Ireland found that above-ground biomass



growth was similar for Italian alder compared to Sitka spruce (Foreman, 2019) which also aligns well
with our observations.

### 3.3 BVOC emissions from tree branches

#### 3.3.1    Italian alder

Italian alder (*Alnus cordata*) emitted very low amounts of isoprene, ranging between <0.0005 –
0.035 µg C $g_{dw}^{-1}$ $h^{-1}$ (standardised 0.017–0.037 µg C $g_{dw}^{-1}$ $h^{-1}$ ) depending on season (Table 4),
comparable with previous standardised emission rates reported as <0.1–3 µg $g_{dw}^{-1}$ $h^{-1}$ (0.09 – 2.64
µg C $g_{dw}^{-1}$ $h^{-1}$) (Calfapietra et al., 2009).
Average measured emissions for total monoterpene ranged between 0.041–0.393 µg C $g_{dw}^{-1}$ $h^{-1}$
(standardised 0.073–1.5 µg C $g_{dw}^{-1}$ $h^{-1}$) with higher emission rates during spring and summer 2018
than in 2019. The major monoterpenes emitted were d-limonene, α-pinene, β-myrcene and β-
pinene, which were consistently emitted through the spring and summer (Figure 1). No previous
data for total or speciated monoterpene emission rates from Italian alder could be found in the
literature. However, other alder species have also been reported to be low emitters of
monoterpenes, and to emit slightly more monoterpenes than isoprene. Studies that report similar
low levels of total monoterpene emissions from alder include 0.8 µg C $g_{dw}^{-1}$ $h^{-1}$ from grey alder
(Hakola et al., 1999), 0.13 µg C $g_{dw}^{-1}$ $h^{-1}$ from black (or common) alder (Aydin et al., 2014) and 1–2 µg
C $g_{dw}^{-1}$ $h^{-1}$ from green alder (*Alnus rugosa*) (Isebrands et al., 1999). For speciated emissions, 3-carene,
β-phellandrene, β-ocimene, p-cymene, sabinene have also been reported to be emitted from *Alder*
*sp*. (Aydin et al., 2014; Copolovici et al., 2014; Hakola et al., 1999; Huber et al., 2000). Emissions of
some monoterpenes such as β-myrcene are suggested to be induced by herbivory by aphids (Blande
et al., 2010). However, since no data on the composition of monoterpenes under laboratory studies
in the absence of herbivory is available for Italian alder it is difficult to know which, if any, of the
monoterpenes measured in our field study may have been induced by previous herbivory.





**Table 4 – Average seasonal BVOC emissions (µg C g⁻¹ h⁻¹) from branches of Sitka spruce, hybrid**
**aspen and Italian alder in SRF plantations, East Grange, Fife, Scotland. Figures in parentheses are**
**standardised deviations.**

|  | Spring 2018 | | | Summer 2018 | | | Autumn 2018 | | | Winter 2019 | | | Spring 2019 | | | Summer 2019 | | |
|---|---|---|---|---|---|---|---|---|---|---|---|---|---|---|---|---|---|---|
|  | Sitka spruce | Hybrid aspen | Italian alder | Sitka spruce | Hybrid aspen | Italian alder | Sitka spruce | Hybrid aspen | Italian alder | Sitka spruce | Hybrid aspen | Italian alder | Sitka spruce | Hybrid aspen | Italian alder | Sitka spruce | Hybrid aspen | Italian alder |
| Days | 4 | 1 | 1 | 2 | 4 | 3 | - | - | - | 3 | - | 2 | 4 | 3 | 4 | 2 | 2 | 4 |
| N | 18 | 5 | 4 | 12 | 18 | 12 | - | - | - | 10 | - | 8 | 10 | 10 | 7 | 7 | 7 | 13 |
| chamber T / ºC | 15.4 | 29.9 | 20.1 | 24.7 | 23.8 | 30.6 | | | | 19.3 | | 16.9 | 25.5 | 23.0 | 22.6 | 30.1 | 29.9 | 26.5 |
|  | (7.3) | (1.4) | (3.1) | (8.9) | (5.6) | (3.0) | | | | (5.2) | | (2.0) | (7.1) | (3.1) | (3.7) | (6.1) | (4.7) | (7.4) |
| PAR / µmol m⁻² s⁻¹ | 607 | 957 | 362 | 662 | 539 | 1018 | | | | 394 | | 298 | 934 | 882 | 1081 | 977 | 957 | 866 |
|  | (464) | (214) | (166) | (530) | (380) | (447) | | | | (217) | | (106) | (481) | (357) | (331) | (609) | (368) | (397) |
| chamber RH / % | 65 | 66 | 82 | 62 | 67 | 39 | | | | 66 | | 74 | 49 | 78 | 61 | 69 | 66 | 59 |
|  | (16) | (2) | (4) | (13) | (17) | (9) | | | | (4) | | (4) | (10) | (17) | (17) | (17) | (6) | (20) |
| Isoprene | 0.365 | 3.091 | 0.010 | 5.904 | 21.115 | 0.035 | | | | 0.031 | | 0.011 | 1.526 | 0.053 | 0.017 | 3.639 | 14.547 | 0.000 |
|  | (0.864) | (0.961) | (0.008) | (3.221) | (17.304) | (0.080) | | | | (0.048) | | (0.000) | (1.887) | (0.038) | (0.020) | (1.872) | (18.616) | (0.014) |
| Standardised Isoprene | 0.688 | 3.163 | 0.060 | 15.046 | 23.487 | 0.037 | | | | 0.139 | | 0.000 | 1.830 | 0.186 | 0.048 | 6.833 | 22.149 | 0.017 |
|  | (1.384) | (0.620) | (0.051) | (8.307) | (11.057) | (0.071) | | | | (0.183) | | (0.000) | (1.725) | (0.130) | (0.064) | (7.013) | (18.159) | (0.043) |
| Total MT | 0.325 | 0.082 | 0.268 | 2.609 | 0.201 | 0.393 | | | | 0.428 | | 0.039 | 1.458 | 0.040 | 0.041 | 2.314 | 0.062 | 0.095 |
|  | (1.045) | (0.042) | (0.114) | (2.888) | (0.251) | (0.340) | | | | (0.902) | | (0.029) | (1.317) | (0.069) | (0.039) | (1.517) | (0.077) | (0.366) |
| Standardised Total MT | 1.949 | 0.090 | 0.711 | 4.534 | 0.259 | 1.503 | | | | 0.665 | | 0.478 | 1.913 | 0.082 | 0.075 | 2.344 | 0.087 | 0.212 |
|  | (7.145) | (0.046) | (0.434) | (4.817) | (0.361) | (2.823) | | | | (1.257) | | (0.406) | (2.220) | (0.103) | (0.073) | (1.652) | (0.069) | (0.720) |
| α-pinene | 0.035 | 0.000 | 0.049 | 0.158 | 0.034 | 0.063 | | | | 0.012 | | 0.019 | 0.026 | 0.009 | 0.013 | 0.189 | 0.006 | 0.047 |
|  | (0.101) | (0.010) | (0.029) | (0.105) | (0.037) | (0.052) | | | | (0.020) | | (0.011) | (0.022) | (0.017) | (0.012) | (0.304) | (0.009) | (0.191) |
| Standardised α-pinene | 0.202 | 0.004 | 0.126 | 0.280 | 0.044 | 0.236 | | | | 0.026 | | 0.070 | 0.036 | 0.024 | 0.024 | 0.221 | 0.011 | 0.106 |
|  | (0.600) | (0.008) | (0.094) | (0.148) | (0.038) | (0.506) | | | | (0.035) | | (0.076) | (0.015) | (0.025) | (0.025) | (0.069) | (0.011) | (0.375) |
| β-pinene | 0.006 | 0.003 | 0.000 | 0.025 | 0.005 | 0.004 | | | | 0.005 | | 0.003 | 0.013 | 0.001 | 0.001 | 0.070 | 0.002 | 0.001 |
|  | (0.018) | (0.002) | (0.001) | (0.017) | (0.006) | (0.007) | | | | (0.008) | | (0.002) | (0.011) | (0.001) | (0.001) | (0.102) | (0.002) | (0.005) |
| Standardised β-pinene | 0.036 | 0.003 | 0.000 | 0.044 | 0.007 | 0.005 | | | | 0.008 | | 0.028 | 0.018 | 0.002 | 0.002 | 0.077 | 0.002 | 0.003 |
|  | (0.0124) | (0.002) | (0.000) | (0.025) | (0.006) | (0.004) | | | | (0.012) | | (0.029) | (0.022) | (0.002) | (0.002) | (1.06) | (0.002) | (0.009) |
| camphene | 0.030 | 0.002 | 0.001 | 0.133 | 0.005 | 0.046 | | | | 0.006 | | 0.001 | 0.010 | 0.000 | 0.000 | 0.040 | 0.000 | 0.001 |
|  | (0.088) | (0.001) | (0.007) | (0.099) | (0.009) | (0.061) | | | | (0.012) | | (0.001) | (0.007) | (0.000) | (0.000) | (0.055) | (0.001) | (0.003) |
| Standardised camphene | 0.175 | 0.002 | 0.006 | 0.237 | 0.008 | 0.058 | | | | 0.019 | | 0.001 | 0.014 | 0.000 | 0.000 | 0.056 | 0.000 | 0.002 |
|  | (0.599) | (0.001) | (0.008) | (0.148) | (0.009) | (0.060) | | | | (0.035) | | (0.003) | (0.015) | (0.001) | (0.000) | (0.068) | (0.001) | (0.006) |
| β-myrcene | 0.174 | 0.025 | 0.02 | 1.772 | 0.010 | 0.149 | | | | 0.264 | | 0.001 | 0.850 | 0.000 | 0.001 | 0.884 | 0.001 | 0.001 |
|  | (0.592) | (0.017) | (0.008) | (2.329) | (0.011) | (0.162) | | | | (0.599) | | (0.001) | (0.806) | (0.001) | (0.001) | (0.425) | (0.002) | (0.003) |
| Standardised β-myrcene | 1.070 | 0.025 | 0.051 | 3.055 | 0.013 | 0.177 | | | | 0.392 | | 0.009 | 1.097 | 0.001 | 0.002 | 0.807 | 0.002 | 0.002 |
|  | (4.052) | (0.0018) | (0.014) | (3.741) | (0.0012) | (0.132) | | | | (0.839) | | (0.003) | (1.256) | (0.002) | (0.003) | (0.279) | (0.002) | (0.006) |

Values shown as 0.000 = <0.0005, - = Not measured

**Table 4 continued.**

|  | Spring 2018 | | | Summer 2018 | | | Autumn 2018 | | | Winter 2019 | | | Spring 2019 | | | Summer 2019 | | |
|---|---|---|---|---|---|---|---|---|---|---|---|---|---|---|---|---|---|---|
|  | Sitka spruce | Hybrid aspen | Italian alder | Sitka spruce | Hybrid aspen | Italian alder | Sitka spruce | Hybrid aspen | Italian alder | Sitka spruce | Hybrid aspen | Italian alder | Sitka spruce | Hybrid aspen | Italian alder | Sitka spruce | Hybrid aspen | Italian alder |
| α-phellandrene | 0.000 | 0.000 | 0.001 | 0.015 | 0.000 | 0.000 | - | - | - | 0.001 | | 0.000 | 0.003 | 0.000 | 0.000 | 0.013 | 0.000 | 0.000 |
|  | (0.000) | (0.000) | (0.001) | (0.012) | (0.000) | (0) | | | | (0.002) | | (0.000) | (0.003) | (0.000) | (0.000) | (0.006) | (0.001) | (0.001) |
| Standardised α-phellandrene | 0.000 | 0.000 | 0.001 | 0.028 | 0.000 | 0.002 | | | | 0.001 | | 0.003 | 0.003 | 0.000 | 0.000 | 0.013 | 0.000 | 0.001 |
|  | (0.000) | (0.000) | (0.002) | (0022) | (0.000) | (0.006) | | | | (0.003) | | (0.004) | (0.003) | (0.000) | (0.000) | (0.006) | (0.001) | (0.002) |
| β-phellandrene | 0.000 | 0.000 | 0.000 | 0.020 | 0.009 | 0.000 | | | | 0.003 | | 0.001 | 0.007 | 0.008 | 0.000 | 0.017 | 0.007 | 0.000 |
|  | (0.000) | (0.000) | (0.000) | (0.011) | (0.011) | (0.00) | | | | (0.006) | | (0.000) | (0.006) | (0.018) | (0.000) | (0.009) | (0.010) | (0.004) |
| Standardised β-phellandrene | 0.000 | 0.000 | 0.000 | 0.035 | 0.008 | 0.000 | | | | 0.004 | | 0.000 | 0.010 | 0.012 | 0.000 | 0.016 | 0.008 | 0.001 |
|  | (0.000) | (0.000) | (0.000) | (0.021) | (0.009) | (0.000) | | | | (0.008) | | (0) | (0.014) | (0.025) | (0.000) | (0.007) | (0.011) | (0.002) |
| d-limonene | 0.078 | 0.047 | 0.160 | 0.426 | 0.108 | 0.092 | | | | 0.120 | | 0.015 | 0.398 | 0.004 | 0.014 | 0.958 | 0.014 | 0.022 |
|  | (0.243) | (0.015) | (0.102) | (0.270) | (0.229) | (0.140) | | | | (0.239) | | (0.011) | (0.351) | (0.009) | (0.015) | (0.886) | (0.017) | (0.062) |
| Standardised d-limonene | 0.460 | 0.048 | 0.426 | 0.748 | 0.143 | 0.876 | | | | 0.185 | | 0.285 | 0.588 | 0.010 | 0.024 | 1.039 | 0.023 | 0.040 |
|  | (1.662) | (0.019) | (0.338) | (0.427) | (0.339) | (1.964) | | | | (0.329) | | (0.255) | (0.837) | (0.020) | (0.024) | (0.987) | (0.015) | (0.123) |
| eucalyptol | 0.001 | 0.007 | 0.004 | 0.053 | 0.012 | 0.016 | | | | 0.014 | | 0.000 | 0.145 | 0.010 | 0.000 | 0.114 | 0.003 | 0.000 |
|  | (0.003) | (0.003) | (0.002) | (0.110) | (0.013) | (0.016) | | | | (0.024) | | (0.020) | (0.384) | (0.023) | (0.001) | (0.088) | (0.04) | (0.001) |
| Standardised eucalyptol | 0.006 | 0.007 | 0.010 | 0.094 | 0.015 | 0.030 | | | | 0.023 | | 0.010 | 0.139 | 0.016 | 0.000 | 0.092 | 0.005 | 0.001 |
|  | (0.002) | (0.003) | (0.006) | (0.056) | (0.015) | (0.042) | | | | (0.037) | | (0.007) | (0.033) | (0.033) | (0.001) | (0.062) | (0.008) | (0.001) |
| 3-carene | 0.000 | 0.000 | 0.035 | 0.008 | 0.017 | 0.023 | | | | 0.003 | | 0.014 | 0.006 | 0.002 | 0.009 | 0.017 | 0.005 | 0.025 |
|  | (0.000) | (0.004) | (0.008) | (0.009) | (0.013) | (0.039) | | | | (0.006) | | (0.003) | (0.006) | (0.003) | (0.013) | (0.015) | (0.007) | (0.101) |
| Standardised 3-carene | 0.000 | 0.001 | 0.090 | 0.013 | 0.021 | 0.118 | | | | 0.006 | | 0.065 | 0.008 | 0.005 | 0.014 | 0.014 | 0.007 | 0.056 |
|  | (0.000) | (0.03) | (0.042) | (0.007) | (0.013) | (0.247) | | | | (0.008) | | (0.062) | (0.008) | (0.003) | (0.017) | (0.009) | (0.006) | (0.198) |
| linalool | 0.000 | 0.000 | 0.000 | 0.000 | 0.000 | 0.000 | | | | 0.000 | | 0.000 | 0.000 | 0.006 | 0.003 | 0.008 | 0.024 | 0.000 |
|  | (0.000) | (0.000) | (0.000) | (0.000) | (0.000) | (0.000) | | | | (0.001) | | (0.000) | (0.001) | (0.010) | (0.005) | (0.006) | (0.030) | (0.000) |
| Standardised linalool | 0.000 | 0.000 | 0.000 | 0.000 | 0.000 | 0.000 | | | | 0.001 | | 0.002 | 0.000 | 0.012 | 0.007 | 0.006 | 0.029 | 0.000 |
|  | (0.000) | (0.000) | (0.000) | (0.000) | (0.000) | (0.000) | | | | (0.001) | | (0.002) | (0.001) | (0.024) | (0.013) | (0.004) | (0.003) | (0.001) |
| γ-terpinene | 0.000 | 0.00 | 0.000 | 0.000 | 0.000 | 0.000 | | | | 0.000 | | 0.000 | 0.000 | 0.000 | 0.000 | 0.004 | 0.000 | 0.000 |
|  | (0.000) | (0.00)0 | (0.000) | (0.000) | (0.000) | (0.000) | | | | (0.000) | | (0.000) | (0.000) | (0.000) | (0.000) | (0.003) | (0.001) | (0.000) |
| Standardised γ-terpinene | 0.000 | 0.000 | 0.000 | 0.000 | 0.000 | 0.000 | | | | 0.000 | | 0.003 | 0.000 | 0.000 | 0.000 | 0.003 | 0.000 | 0.000 |
|  | (0.000) | (0.000) | (0.000) | (0.000) | (0.000) | (0.000) | | | | (0.000) | | (0.005) | (0.000) | (0.000) | (0.001) | (0.002) | (0.001) | (0.001) |

Values 0.000 = <0.0005, - = Not measured



### 3.3.2 Hybrid aspen

Isoprene emissions from hybrid aspen ranged from 0.053 to 21 µg C $g_{dw}^{-1}$ $h^{-1}$ (standardised 0.19–23 µg C $g_{dw}^{-1}$ $h^{-1}$) (Table 4). No measurements were made during autumn senescence or in winter on the bare branches. Emissions were lower in spring for the newly emerged leaves compared to summer (Figure 1). As noted in Section 3.1, the onset of spring at the field site was earlier in 2018 compared to 2019. European aspen (*Populus tremula*) measured in late spring (May) two weeks after bud burst has also previously been reported to have a lower emission rate than in summer (Hakola et al., 1998). Isoprene emission rates made on leaves (not branches) on aspen in spring in the boreal forest were also reported to be a third of the emission rate measured in the middle of summer (Fuentes et al., 1999). In our study, the hybrid aspen plantation showed signs of stress thought to be associated with lower rainfall and soil moisture locally during summer 2018 causing a yellowing of leaves and early leaf shedding in July. It is widely accepted that isoprene emissions increase with increases in temperature and PAR (Guenther et al., 1991; Monson and Fall, 1989) but that under stress during drought, isoprene can be emitted at much higher rates than usual, only to eventually decline as resources are depleted in the leaves (Brilli et al., 2007; Seco et al., 2015). However, standardised isoprene emissions measured during this study on green aspen leaves did not differ between the two years, 2018 (23 µg C $g_{dw}^{-1}$ $h^{-1}$) and 2019 (22 µg C $g_{dw}^{-1}$ $h^{-1}$) despite the signs of stress in 2018 noted above. The standardised isoprene emissions for hybrid aspen reported here were much lower than those previously reported for European aspen, 51 µg $g_{dw}^{-1}$ $h^{-1}$ (i.e. 45 µg C $g_{dw}^{-1}$ $h^{-1}$) (Hakola et al., 1998).

Total monoterpene emissions for hybrid aspen ranged from 0.040 - 0.20 µg C $g_{dw}^{-1}$ $h^{-1}$ (standardised 0.082 - 0.259 µg C $g_{dw}^{-1}$ $h^{-1}$) with substantially higher emissions occurring in summer 2018 (Table 4, Figure 1). Increased emissions for some monoterpenes have been shown to be predominately driven by increases in temperature (Guenther et al., 1991). In particular d-limonene, the major





monoterpene emitted here, was found to correlate with an increase in temperature, comparable to
elevated temperature experiments for European aspen (Hartikainen et al., 2009). However, total
monoterpene emission rates were an order of magnitude lower in summer during our study, closer
to the findings of Brilli et al. (2014) from a SRC plantation of poplar, and in contrast to the 4.6 µg $g_{dw}^{-}$
$^{1}$ h$^{-1}$ (4.1 µg C $g_{dw}^{-1}$ h$^{-1}$) reported for European aspen by Hakola et al. (1998). D-limonene, $\alpha$-pinene,
carene and $\beta$-phellandrene collectively accounted for 50–95% of the total monoterpene emissions,
although the composition for different days was highly variable (Figure 1). Emissions of $\alpha$-
phellandrene peaked at 27% of total monoterpenes in April when catkins were present but were
otherwise < 13% (except on 6 June 2018).

Previously studies on European aspen report monoterpene emissions of 3-carene, limonene, $\alpha$-
pinene, trans-ocimene, eucalyptol, $\beta$-myrcene and sabinene (Aydin et al., 2014; Hakola et al., 1998;
Hartikainen et al., 2009) and on hybrid aspen (*Populus tremula – Populus tremuloides*) report $\alpha$-
pinene, $\beta$-pinene and $\beta$-ocimene, (Blande et al., 2007), although differences between clones were
noted.



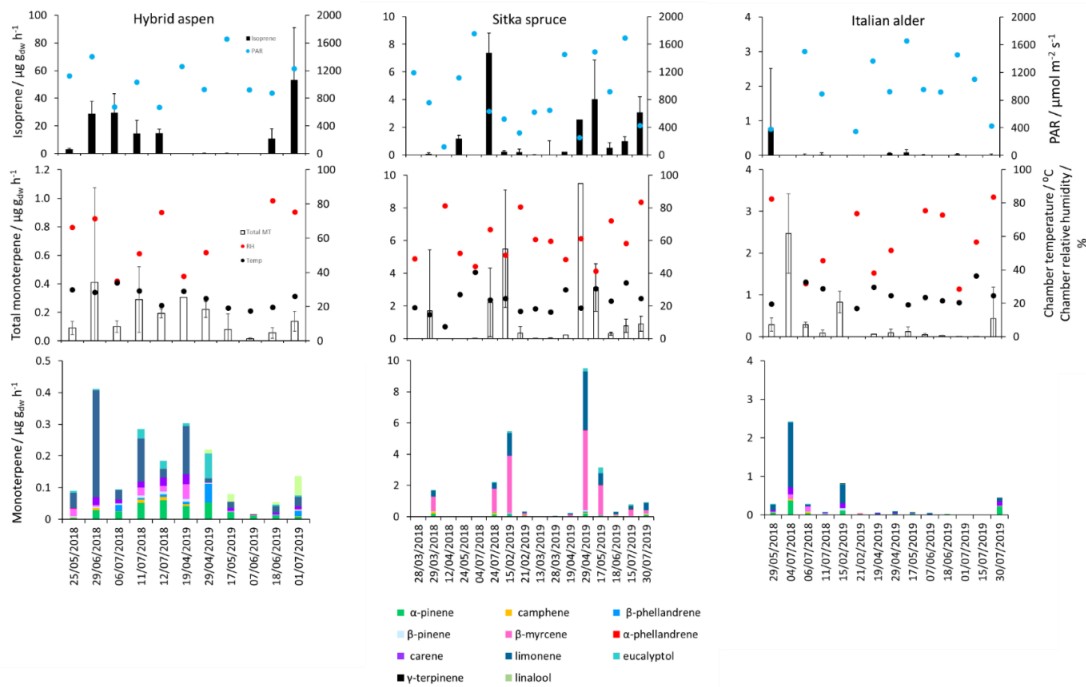


**Figure 1 – Average isoprene, total monoterpene and speciated monoterpene emissions from branches of hybrid aspen, Italian alder and Sitka spruce trees in SRF plantations at the East Grange site, Fife, between March 2018 and July 2019. Error bars show standard deviation of all measurements made on a given day. Blue, red and black circles show average PAR, chamber relative humidity and temperature, respectively. Note that emission scales differ between tree species**

### 3.3.3 Sitka spruce

Average isoprene emissions from Sitka spruce ranged from 0.031 µg C $g_{dw}^{-1}$ $h^{-1}$ (standardised 0.14 µg C $g_{dw}^{-1}$ $h^{-1}$) in winter to 5.9 µg C $g_{dw}^{-1}$ $h^{-1}$ (standardised 15.0 µg C $g_{dw}^{-1}$ $h^{-1}$) in summer (Table 4), which are comparable to the range of previously reported emissions from UK field measurements, 0.005-1.48 µg $g_{dw}^{-1}$ $h^{-1}$ (standardised 0.88–14.1 µg C $g_{dw}^{-1}$ $h^{-1}$) (Street et al., 1996). Standardised isoprene emissions were lower in spring than summer during both years in our study (Figure 1). Standardised isoprene emissions in summer 2018 (15.0 µg C $g_{dw}^{-1}$ $h^{-1}$) were more than twice those in summer 2019 (6.8 µg C $g_{dw}^{-1}$ $h^{-1}$), likely reflective of the wetter and cooler conditions in 2019. However, laboratory measurements using trees acclimatised at a constant laboratory temperature of 20 °C and PAR of





1000 µmol m$^{-2}$ s$^{-1}$ for a week prior to sampling showed emission rates similar to summer 2018
emission rates, 13.4 µg g$_{dw}$$^{-1}$ h$^{-1}$ (11.8 µg C g$_{dw}$$^{-1}$ h$^{-1}$)  (Hayward et al, 2004). The isoprene emissions in
our study declined dramatically at higher chamber temperatures, > 31 $^o$C , despite the high PAR
levels. An optimum of 33 $^o$C for isoprene emissions from Sitka spruce was noted by Street et al.
(1996), although a higher optimum of 39 $^o$C was suggested by Hayward et al. (2004) based on a
laboratory study. We therefore suggest that Sitka spruce trees acclimatised under field conditions in
Scotland with variable day and night temperatures and light levels, may have a lower optimum
temperature than observed under laboratory conditions.  The previous suggestion that Sitka spruce
reaches maximum emissions of isoprene at a low level of PAR of 300 µmol m$^{-2}$ s$^{-1}$ (Hayward et al.,
2004) was difficult to confirm under field conditions as high PAR values were correlated with high
temperatures (Figure 2). However, it is worth noting that the majority of field emissions collected by
Street et al. (1996) align well with the emissions measured at lower PAR and temperature in this
study (Figure 2).

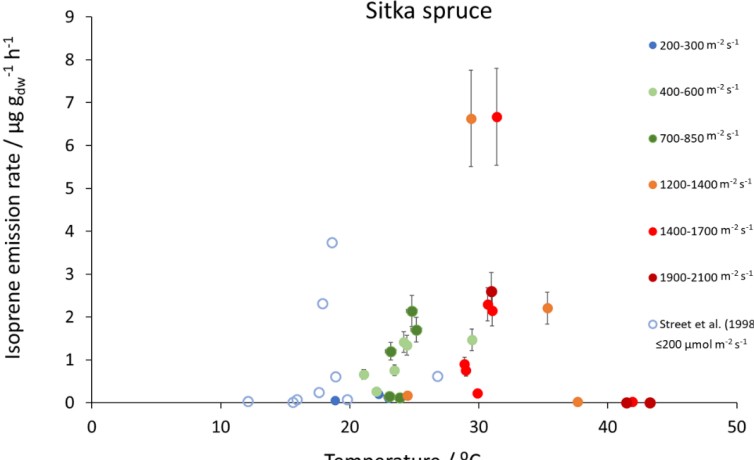


**Figure 2 – Isoprene emissions as a function of PAR and temperature for Sitka spruce at East Grange**
**SRF site and from Street et al. (1996) at PAR ≤ 200 µmol m$^{-2}$ s$^{-1}$.**





Total monoterpene emissions from Sitka spruce peaked on the 29th April 2019 (9.5 µg C $g_{dw}^{-1}$ $h^{-1}$)
coinciding with the new shoot extension growth on the branches (Figure 1). Monoterpene emissions
have shown to be present in spring in advance of isoprene emissions for Norway spruce (*Picea abies*)
(Hakola et al., 2003). Overall, monoterpene emissions were generally higher in summer than in
spring (Table 4). Total monoterpene emissions were still higher in 2018 (standardised 4.5 µg C $g_{dw}^{-1}$
$h^{-1}$) than in 2019 (2.3 µg C $g_{dw}^{-1}$ $h^{-1}$) even once standardised to 30 °C, which could indicate an
increased release of monoterpenes in response to the drier warmer conditions. The total
monoterpene emissions  in 2019 are comparable to the previously reported total monoterpene
emission of 3.0 µg $g_{dw}^{-1}$ $h^{-1}$ (2.6 µg C $g_{dw}^{-1}$ $h^{-1}$) from a laboratory study (Hayward et al., 2004).
Monoterpene emissions from Sitka spruce comprised predominately of β-myrcene, d-limonene, α-
pinene and eucalyptol, collectively accounting for 83–97% of total monoterpenes across all
measurement days (Figure 1).

β-myrcene was the most abundant, consistent with the findings of Geron et al. (2000), and has been
reported to be highest during spring in leaf oils, associated with new growth in this species, only to
decline later in  the growing season (Hrutfiord et al., 1974) but this was not evident during our study.
d-limonene emission rates reported during our study are comparable in size to Hayward et al.
(2004), although not the dominant monoterpene as previously reported. Furthermore, other studies
have also reported limonene to be present in smaller quantities than α-pinene and β-myrcene
(Beverland et al., 1996; Hrutfiord et al., 1974). Monoterpene composition was generally consistent
between measurements throughout our study even though different branches and trees were
measured, which is perhaps a consequence of growing plantation trees propagated vegetatively
rather than by seed. However, the variability between the previous literature discussed here may
point towards the potential for different chemotypes within Sitka spruce, as suggested by (Forrest,
2011) and similar to that of Norway spruce (Hakola et al., 2017) and Scots pine (Bäck et al., 2012).





Given the dominance of Sitka spruce plantations in the UK (and Ireland), the potential for variation
within this species, and the limited literature data on BVOC emissions, we suggest further
measurements are needed at the branch and canopy level to fully assess the monoterpene
composition and subsequent impact on air quality.

### 3.4  BVOC emissions from forest floor
The forest floor has been reported as both a source of BVOCs (Asensio et al., 2007a, 2007b;
Bourtsoukidis et al., 2018; Greenberg et al., 2012; Hayward et al., 2001; Insam and Seewald, 2010;
Janson, 1993; Leff and Fierer, 2008; Mäki et al., 2019a; Peñuelas et al., 2014) and a sink, particularly
for isoprene (Cleveland and Yavitt, 1997, 1998; Owen et al., 2007; Trowbridge et al., 2020). Leaf litter
is a known source of forest floor BVOCs (Gray et al., 2010; Greenberg et al., 2012; Isidorov and
Jdanova, 2012). Data discussed here are the net flux of the opposing processes of source and sink.
Monoterpene emissions from the forest floor (Hayward et al., 2001) have previously been
standardised using G93 (Eq. (3)) on the assumption that air temperature is the main driver of
emissions of monoterpenes. However, these algorithms are based on empirical data and were not
designed to normalise negative emissions (uptake). In addition, what drives the sources and sinks of
the forest floor is often more complex; and although some models have been developed from
laboratory or field studies for litter, soils and the forest floor (Greenberg et al., 2012; Mäki et al.,
2017, 2019b) the models may be difficult to apply outside of the studies in which they were
developed. A process-based model applicable to a range of forest floor types is still lacking (Tang et
al., 2019). We therefore did not standardise the BVOC emissions from the forest floor and present
only measured fluxes in this section.

The total monoterpene emissions from the forest floor were highly variable between the three
chambers within the plots as demonstrated by a relative standard deviation range of 35% to 170%





for a given day, illustrating the highly heterogeneous soil and litter environment. All chamber
measurements made on the same day were averaged per species, and presented as a single flux
value (Figure 3) and then grouped according to season and year (Table 5).

### 3.4.1 Italian alder

Negative fluxes for total monoterpenes were measured on two occasions, 4th July and 24th July. The
highest total monoterpene emissions were observed on the 18th October 2018 (18 µg C m$^{-2}$ h$^{-1}$) and
7th June 2019 (24 µg C m$^{-2}$ h$^{-1}$) (Figure 3). Day to day variations were associated to some degree with
changes in chamber temperature and soil moisture (Figure 3). Seasonal variations in average
emissions were also apparent (Table 5). The forest floor acted as a sink for monoterpenes during
summer 2018 when there was bare soil inside the collars. During summer 2019 vegetation grew
inside the soil collars and resulted in the forest floor being a more substantial source of
monoterpenes (Figure 4). Monoterpene composition reflected the seasonal changes that occurred
on the forest floor. The monoterpenes emitted in autumn (October 2018) were dominated by d-
limonene, α-pinene and 3-carene and some β-myrcene, consistent with the composition of Italian
alder foliage and attributed to the accumulation of leaf litter. However, the profile in June 2019
during the highest total monoterpene emissions showed significant emissions of γ-terpinene and α-
phellandrene and likely reflects the changing understorey vegetation, hogweed sp., growing inside
the chamber collars and which was only present in the alder plantations. The particular species at
East Grange was not identified but *Heracleum mantegazzianum* (giant hogweed) has been
determined to be a substantial γ-terpinene emitter (Matoušková et al., 2019). This highlights the
importance of the specific understorey vegetation to the overall monoterpene flux composition.





### 3.4.2 Hybrid aspen

The highest total monoterpene emissions, 9.18 µg C m$^{-2}$ h$^{-1}$ and 5.83 µg C m$^{-2}$ h$^{-1}$, occurred in July 2018 and were associated with the lowest soil moisture and warm temperatures. In contrast, negative monoterpene emissions were also observed in July (24$^{th}$) and seem to be associated with an increase in soil moisture (Figure 3). Overall spring (0.30 µg C m$^{-2}$ h$^{-1}$) and summer (0.06 µg C m$^{-2}$ h$^{-1}$) total monoterpene emission rates in 2019 (Table 5 ) were smaller by an order of magnitude than in spring (0.71 µg C m$^{-2}$ h$^{-1}$) and summer (3.84 µg C m$^{-2}$ h$^{-1}$) 2018. Higher rainfall during 2019 (Supplementary Information S1) resulted in increased soil moisture (Figure 3) which may have suppressed some monoterpene emissions (Asensio et al., 2007b). In addition, during 2018, litterfall started in July and peaked in October by which time the canopy had lost all its leaves.

The composition of the monoterpene emissions from the forest floor during 2018 was similar to those measured from the branch chambers (Figure 1) and was consistent between days. The main monoterpenes comprised α-pinene, β-pinene, camphene, d-limonene and 3-carene. The contribution from the floor of an aspen plantation has not previously been investigated, although soils taken from underneath *Populus tremula* trees showed d-limonene as the predominant monoterpene with a maximum emission of 15.9 µg C m$^{-2}$ h$^{-1}$ under laboratory conditions (Owen et al., 2007). Quantifiable emissions of monoterpene from the leaf litter of aspen (*Populus tremuloides*) exist (Gray et al., 2010) although not chemically speciated



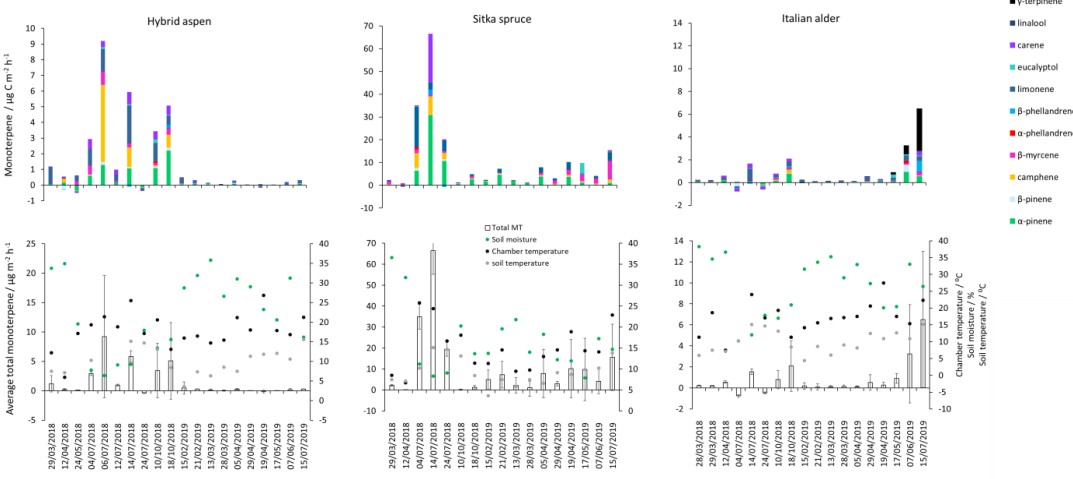


**Figure 3 – Daily average forest floor total monoterpene emissions from Sitka spruce, hybrid aspen**
**and Italian alder SRF plots at East Grange, Fife during 2018-2019. Error bars represent the standard**
**deviation of three forest floor chamber measurements. Green circles are volumetric soil moisture**
**(%), black circles are chamber temperature (°C) and grey circles are soil temperature (°C). Note**
**that emission scales differ between tree species plots.**
### 3.4.3   Sitka spruce

Total monoterpene emissions from the Sitka spruce forest floor peaked during July 2018 (66.5 µg C
$m^{-2}$ $h^{-1}$) and coincided with the highest chamber temperatures and the lowest soil moisture readings
(Figure 3). The lowest emissions (0.03 µg C $m^{-2}$ $h^{-1}$) were observed on the 12th April 2018 when the
temperature was lowest (7.5 °C, Figure 3) suggesting soil moisture and temperature are likely
interacting controlling variables of monoterpene emissions. In addition, there were clear seasonal
differences when measurement days were grouped. Average summertime emissions of total
monoterpenes from the forest floor in 2018 were larger than those measured in 2019 (Table 5).
Temperatures measured in the chambers were 3 °C degrees higher on average during 2018
compared to 2019 which could have contributed to the higher observed emissions although soil
moisture at 7 cm depth was not significantly different. The young Sitka spruce plantation had litter
present all year round unlike in the deciduous species plantations,  but the covering was sparse



(Figure 4) compared to a mature plantation. Total monoterpene emissions in summer 2018 (40.3 µg
C m$^{-2}$ h$^{-1}$) were slightly higher but similar in magnitude to the 33.6  µg m$^{-2}$ h$^{-1}$ (29.6 µg C m$^{-2}$ h$^{-1}$)
previously reported for the upper-most layers of the floor in a mature Sitka spruce plantation
(Hayward et al., 2001). Norway spruce plantation have also been reported to have a slightly higher
emission rate at 50 µg C m$^{-2}$ h$^{-1}$ (Janson et al., 1999).

The monoterpene composition profile in 2018 was comparable to 2019 and consistent with the
branch emissions recorded during our study, the major emitted monoterpenes being β-myrcene, α-
pinene, β-pinene, d-limonene and camphene. β-myrcene accounted for a larger percentage, 20–
50%, of emissions in summer 2019 compared to only 5–10% in summer 2018 (Table 5), although
there is no obvious explanation for this difference.




Figure 4 – Changes in the presence of leaf litter, herbaceous plants and grasses inside the forest floor chambers of (a) Italian alder (b) hybrid aspen and (c) Sitka spruce SRF plots at East Grange, Fife during 2019.





**Table 5 – Seasonal variation in forest floor emissions (µg C m$^{-2}$ h$^{-1}$) of monoterpenes from Sitka**
**spruce, hybrid aspen and Italian alder SRF plots, at East Grange, Fife, Scotland, in 2018–19.**

| | Spring 2018 | | | Summer 2018 | | | Autumn 2018 | | | Winter 2019 | | | Spring 2019 | | | Summer 2019 | | |
|---|---|---|---|---|---|---|---|---|---|---|---|---|---|---|---|---|---|---|
| Plantation type | Sitka spruce | Hybrid aspen | Italian alder | Sitka spruce | Hybrid aspen | Italian alder | Sitka spruce | Hybrid aspen | Italian alder | Sitka spruce | Hybrid aspen | Italian alder | Sitka spruce | Hybrid aspen | Italian alder | Sitka spruce | Hybrid aspen | Italian alder |
| Days | 2 | 2 | 3 | 3 | 6 | 3 | 2 | 2 | 2 | 3 | 3 | 3 | 6 | 18 | 6 | 1 | 1 | 1 |
| N | 2 | 4 | 4 | 3 | 8 | 3 | 2 | 4 | 4 | 9 | 9 | 9 | 17 | 18 | 17 | 1 | 1 | 2 |
| air T / °C | 7.6 (1.3) | 9.0 (3.6) | 11.2 (5.2) | 21.1 (4.5) | 19.6 (4.1) | 18.5 (4.2) | 14.8 (4.7) | 16.3 (4.3) | 15.5 (3.7) | 12.6 (1.1) | 12.4 (1.5) | 13.5 (0.5) | 13.9 (2.0) | 16.4 (2.4) | 16.0 (3.8) | 22.5 (0.0) | 16.0 | 20.6 (0.0) |
| chamber T / °C | 7.6 (1.3) | 9.0 (3.6) | 11.2 (5.2) | 21.2 (4.2) | 20.0 (4.2) | 20.6 (4.9) | 14.4 (4.2) | 16.8 (4.4) | 15.4 (4.6) | 11.8 (2.3) | 15.7 (1.5) | 15.5 (1.3) | 13.8 (2.8) | 19.3 (4.0) | 19.5 (4.2) | 22.9 (0.7) | 21.2 | 22.3 (0.0) |
| soil T / °C | 5.3 (1.1) | 6 (1) | 6.9 (0.7) | 14.3 (0.2) | 14.3 (0.9) | 13.4 (2.7) | 9.8 (2.5) | 10.6 (1.9) | 10.8 (2.7) | 6.2 (1.1) | 5.7 (1.7) | 6.4 (1.8) | 8.5 (1.4) | 10.3 (1.8) | 10.7 (1.8) | 13.8 (0.0) | 15.6 | 15.2 (0.0) |
| chamber RH / % | - | - | - | - | - | - | - | - | - | 88 (6) | 81.4 (4.5) | 77 (3) | 74 (9) | 73 (8) | 88 (11) | 70 (7) | 78 | 79 (0) |
| soil moisture / % | 34 (3) | 36 (2) | 37 (2) | 20 (8.0) | 12 (5) | 13.4 (4.0) | 14 (3) | 14 (2) | 19.0 (2.3) | 21 (3) | 32.2 (3.6) | 34 (3) | 14 (2) | 27 (4) | 27 (6) | 15 (1) | 31 | 26 (0) |
| α-pinene | -0.067 (0.372) | 0.113 (0.075) | 0.119 (0.111) | 15.954 (13.059) | 0.557 (0.736) | -0.050 (0.135) | 1.627 (1.443) | 1.634 (1.991) | 0.454 (0.708) | 2.661 (3.225) | 0.230 (0.522) | 0.020 (0.069) | 2.167 (3.624) | 0.005 (0.064) | 0.156 (0.459) | 1.067 (1.18) | 0.112 | 0.557 (0.187) |
| β-pinene | 0.052 (0.034) | -0.150 (0.176) | -0.019 (0.023) | 0.724 (0.579) | 0.076 (0.114) | 0.042 (0.165) | 0.086 (0.010) | 0.145 (0.166) | 0.042 (0.038) | 0.209 (0.271) | 0.054 (0.111) | 0.002 (0.007) | 0.224 (0.387) | 0.007 (0.023) | 0.084 (0.305) | 0.217 (0.191) | 0.004 | 0.037 (0.003) |
| Camphene | 0.130 (0.112) | 0.126 (0.234) | 0.013 (0.004) | 5.775 (2.692) | 1.386 (3.408) | -0.011 (0.038) | 0.255 (0.174) | 0.456 (0.784) | 0.191 (0.275) | 0.142 (0.235) | 0.213 (0.634) | 0.000 (0.008) | 0.687 (1.578) | 0.000 (0.004) | 0.010 (0.022) | 1.248 (1.453) | 0.000 | 0.000 (0.000) |
| β-myrcene | 0.930 (0.447) | 0.014 (0.015) | 0.009 (0.012) | 1.046 (0.533) | 0.426 (0.540) | 0.024 (0.045) | 0.521 (0.483) | 0.272 (0.339) | 0.172 (0.139) | 0.115 (0.256) | 1.255 (3.761) | 0.011 (0.028) | 4.839 (13.585) | 0.005 (0.011) | 0.034 (0.075) | 8.145 (8.828) | 0.002 | 0.270 (0.020) |
| α-phellandrene | 0.006 (0.006) | 0.004 (0.005) | 0.000 (0.003) | 0.355 (0.636) | 0.009 (0.012) | 0.002 (0.002) | 0.000 (0.002) | 0.064 (0.106) | 0.002 (0.007) | 0.011 (0.015) | 0.025 (0.073) | 0.000 (0.000) | 0.055 (0.145) | 0.000 (0.001) | 0.027 (0.107) | 0.118 (0.167) | 0.000 | 0.075 (0.106) |
| β-phellandrene | 0.000 (0.000) | -0.002 (0.003) | 0.000 (0.000) | 0.481 (1.669) | -0.020 (0.037) | -0.021 (0.058) | 0.005 (0.006) | 0.125 (0.226) | 0.085 (0.120) | 0.020 (0.035) | 0.010 (0.028) | 0.000 (0.000) | 0.031 (0.092) | 0.000 (0.000) | 0.003 (0.013) | 0.152 (0.112) | 0.003 | 0.965 (1.290) |
| d-limonene | 0.263 (0.391) | 0.566 (1.014) | 0.167 (0.078) | 8.417 (8.037) | 0.997 (0.888) | 0.270 (0.679) | 0.428 (0.373) | 0.860 (0.933) | 0.260 (0.199) | 0.767 (0.983) | 0.640 (1.450) | 0.095 (0.210) | 2.386 (5.456) | 0.038 (0.053) | 0.192 (0.298) | 3.505 (3.375) | 0.087 | 0.400 (0.021) |
| Eucalyptol | 0.003 (0.002) | 0.002 (0.002) | 0.004 (0.011) | 0.087 (0.160) | 0.040 (0.088) | -0.025 (0.052) | 0.133 (0.132) | 0.150 (0.187) | -0.002 (0.007) | 0.006 (0.011) | 0.053 (0.144) | 0.002 (0.004) | 0.851 (2.980) | 0.000 (0.003) | 0.077 (0.152) | 0.342 (0.346) | 0.015 | 0.065 (0.007) |
| 3-carene | -0.189 (0.276) | 0.034 (0.032) | 0.093 (0.125) | 7.446 (12.140) | 0.372 (0.496) | 0.035 (0.335) | 0.086 (0.006) | 0.552 (0.621) | 0.228 (0.233) | 0.020 (0.029) | 0.055 (0.063) | 0.003 (0.054) | 0.077 (0.147) | 0.001 (0.066) | 0.016 (0.047) | 0.564 (0.077) | 0.049 | 0.347 (0.066) |
| Linalool | 0.000 (0.000) | 0.000 (0.000) | 0.000 (0.000) | 0.000 (0.000) | 0.000 (0.000) | 0.000 (0.000) | 0.000 (0.000) | 0.000 (0.000) | 0.000 (0.000) | 0.001 (0.002) | 0.005 (0.013) | 0.000 (0.001) | -0.000 (0.002) | 0.001 (0.002) | 0.001 (0.004) | 0.012 (0.003) | 0.016 | 0.080 (0.007) |
| γ-terpinene | 0.000 (0.000) | 0.000 (0.000) | 0.000 (0.000) | 0.000 (0.000) | 0.000 (0.000) | 0.000 (0.000) | 0.000 (0.000) | 0.000 (0.000) | 0.000 (0.000) | 0.001 (0.002) | 0.003 (0.003) | 0.000 (0.001) | 0.011 (0.037) | 0.000 (0.002) | 0.128 (0.386) | 0.157 (0.215) | 0.007 | 3.709 (5.187) |
| Total MT | 1.128 (1.559) | 0.707 (0.977) | 0.387 (0.210) | 40.286 (23.999) | 3.843 (5.490) | 0.111 (1.254) | 3.141 (2.615) | 4.257 (4.706) | 1.433 (1.664) | 3.954 (4.970) | 2.543 (6.737) | 0.135 (0.225) | 11.330 (24.084) | 0.057 (0.174) | 0.729 (1.567) | 15.527 (15.797) | 0.296 | 6.506 (6.488) |

T = Temperature, N = Number of measurements, - = Not measured, RH = Relative humidity, 0.000 =
values <0.0005

### 3.5 Plantation-scale isoprene and total monoterpene emissions


#### 3.5.1 Relative contribution of forest floor and canopy emissions

Forest floor and branch emissions were sometimes measured on the same occasion enabling
calculation of the contribution of each source to the total monoterpene emissions of the plantation
per square metre of ground (based on non-standardised data) (Figure 5). In most cases, particularly
in summer, emissions from the canopy dominated. For Sitka spruce, high monoterpene emissions
from the plantation occurred when canopy emissions were high which supports previous
summertime observations on conifer sp. that the forest floor contributes little to the overall forest
monoterpene emissions (Hayward et al., 2001; Janson, 1993). We found that in some instances,
more often in spring when canopy foliage was sparse (alder and aspen) or dormant due to cold





temperatures (spruce), the forest floor contributed the majority of the plantation monoterpene
emissions. This trend was also reported for conifer sp. in the boreal forest (Mäki et al., 2019b).

For hybrid aspen the opposite was true with the forest floor contributing more in the summer, as a
result of understorey vegetation or early litter fall, contributing up to 40% of the total monoterpene
emissions of the plantation. In the Italian alder plantation the contribution was more mixed. Canopy
emissions in late winter/ early spring were only from the alder flowers (catkins). The low observed
emissions at this time of year from the forest floor were likely caused by colder temperatures and
high soil moisture. However, later in spring (April) monoterpene emissions came largely from the
forest floor (90%) as understorey vegetation began to grow and soil temperatures also increased.
The canopy at this point was at the stage of leaf emergence when the foliage was sparse and so
contributed little to the overall emissions. However, by summer just over half of the monoterpenes
came from the canopy (now in full foliage) and the forest floor contributed around 40% of the
monoterpenes, related to the presence of understorey vegetation.




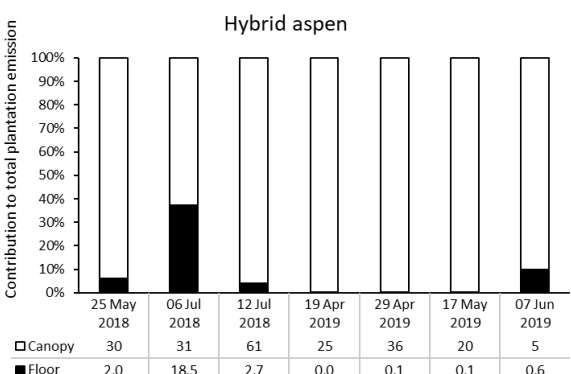

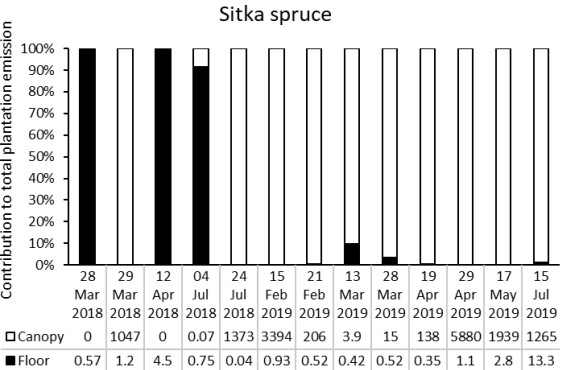

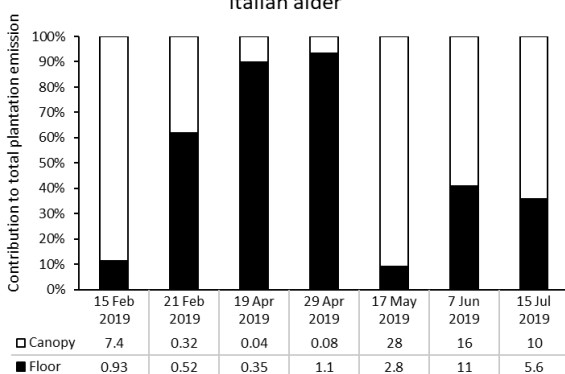

**Figure 5 – Percentage contribution of canopy (white bar) and forest floor (black bar) emissions to the total monoterpene emissions from SRF plantations at East Grange, Fife, Scotland. Numbers below the bars are the total monoterpene emissions in μg C m$^{-2}$ h$^{-1}$.**



### 3.5.2 Modelled above-canopy fluxes


This section discusses modelled emissions of BVOC from the canopy per $m^2$ of ground. The "bottom
up" approach of estimating BVOC emissions in this study using the chamber technique is useful for
determining the contribution of different ecosystem components to BVOC emissions, but in this
section emissions do not include modelled forest floor emissions. It is noted that forest floor
processes are still being integrated into models in order to reliably capture the full complexity of the
forest floor BVOC emissions for prediction purposes (Tang et al., 2019).

Average standardised summertime emission factors for each tree species in section 3.3  (derived
using the simplified G93 algorithms) (Table 3) were adjusted on an hourly basis by the Pocket
MEGAN 2.1 excel beta 3 calculator to derive hourly BVOC emissions per unit ground area (Guenther
et al., 2012). This allowed for a more advanced method of estimation of monthly and subsequent
annual BVOC emissions from the canopy across two years (2018–2019) and two locations, East
Grange (Scotland) and Alice Holt (England) for a given air temperature, PAR and the influence of
these parameters over the previous 24 and 240 hours. In addition, changing LAI across the year
(Table 2) had an influence on the biomass density of the canopy which influenced the emission rate
of BVOCs per unit area of ground. Similar to previous modelling studies (Ashworth et al., 2015;
Zenone et al., 2016) standardised average summertime measurements were used as the basis for
this calculation.

Given the above, modelled average diurnal canopy emissions of isoprene for hybrid aspen were
calculated to be approximately 2 mg C $m^{-2}_{ground}$ $h^{-1}$, rising to a maximum of 7 mg C $m^{-2}_{ground}$ $h^{-1}$ in July,
the warmest month, across both years (Figure 6A). These modelled emissions for the UK are broadly
comparable to those reported from measured eddy covariance flux measurements above a
hardwood forest, comprising primarily of aspen (*Populus tremuloides* and *Populus grandidentata*,





LAI: 3.24-3.75) in Michigan USA and the boreal forest in Canada (predominantly *Populus tremuloides,*
LAI: 2.4) where the average summertime emissions are reported to peak at 11 mg C m$^{-2}_{ground}$ h$^{-1}$ and
6.87 mg C m$^{-2}_{ground}$ h$^{-1}$ respectively (Fuentes et al., 1999; Pressley et al., 2006).

Average total monoterpene emissions are two orders of magnitude smaller than isoprene (Figure
6B) for hybrid aspen. Figure 6 (C and D)) highlights the difference in the relative magnitudes of
emissions between the three SRF species. Average emissions from the canopy of Italian alder for
isoprene (0.002 mg C m$^{-2}_{ground}$ h$^{-1}$) and monoterpene (0.05 mg C m$^{-2}_{ground}$ h$^{-1}$) were very small and no
above-canopy measurements could be found in the literature for comparison. For Sitka spruce
average canopy scale emissions for July in Scotland were modelled to be 1.5 mg C m$^{-2}_{ground}$ h$^{-1}$ and 0.5
mg C m$^{-2}_{ground}$ h$^{-1}$ for isoprene and total monoterpene respectively. There has only been one attempt
in the UK to quantify BVOC directly above a Sitka spruce plantation (Beverland et al., 1996) where a
relaxed eddy accumulation system was used and average isoprene emissions were reported to be
0.146 mg C m$^{-2}_{ground}$ h$^{-1}$ in a 24-h period in early July (temperature range 7-19 °C).






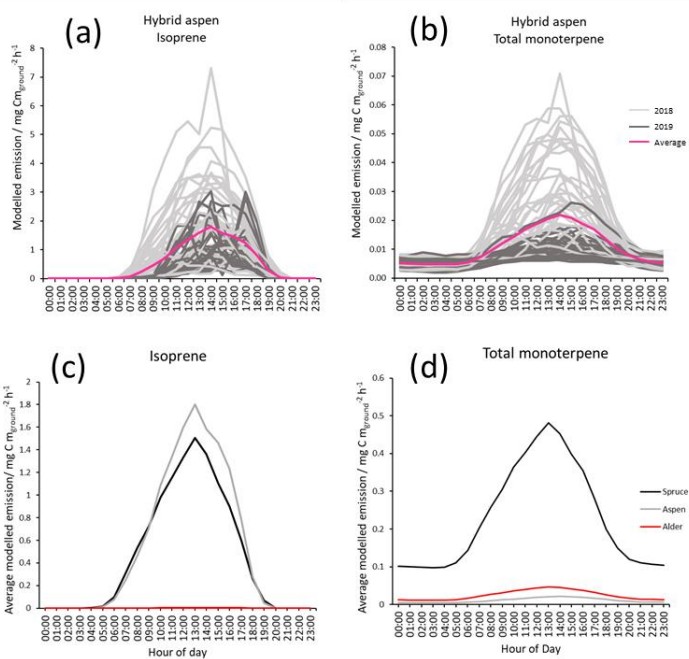


**Figure 6 – Modelled diurnal canopy emissions for July using MEGAN 2.1 of (a) isoprene from**
**hybrid aspen 2018 (light grey), 2019 (dark grey) and combined average emission rate (pink), (b)**
**total monoterpene hybrid aspen (light grey), 2019 (dark grey) and combined average emission rate**
**(pink), (c) average modelled isoprene for three SRF species, spruce (Black), aspen (grey) and alder**
**(red) for July 2018–2109, (d) average modelled total monoterpene for three SRF species, spruce**
**(Black), aspen (grey) and alder (red) for July 2018–2109. Results used measured PAR, temperature**
**and the average summer branch emission potentials collected during this study (Table 3).**

739

740

### 3.5.3    Annual above-canopy fluxes per hectare for a UK planation

Table 6 shows the modelled annual BVOC emissions per hectare of plantation for each species for

the two meteorological years (2018-2019) at East Grange (EG) in Scotland, and for the

contemporaneous meteorology experienced in southern England (at Alice Holt (AH)). The modelled

annual fluxes of isoprene and total monoterpenes per hectare of Sitka spruce plantation averaged

over the two contrasting years were roughly similar, at 13.8 and 15.7 kg C ha$^{-1}$ y$^{-1}$, respectively.

Hybrid aspen was modelled to emit only an average of 0.3 kg C ha$^{-1}$ y$^{-1}$ total monoterpene but much





more isoprene (15.5 kg C ha$^{-1}$ y$^{-1}$), whereas the model estimated that Italian alder emitted minimal
isoprene (0.02 kg C ha$^{-1}$ y$^{-1}$ on average) but larger monoterpene emissions  of 0.81 kg C ha$^{-1}$ y$^{-1}$.

It is worth noting that use of an average summer flux could lead to a potential overestimation of
emissions during other seasons and the subsequent total annual flux. Modelled isoprene emissions
from Sitka spruce during 2018 for both EG and AH were higher than monoterpene emissions. In
2019, however, monoterpene emissions were more abundant than isoprene emissions using the EG
meteorology data and of the same magnitude using the AH meteorology data. The lower PAR during
2019, which was more pronounced for EG than AH, limited the isoprene emissions. Monoterpenes
were less affected as these were only temperature driven. The relative proportions of isoprene and
monoterpenes in the atmosphere are important since they have differing effects on the formation
and concentration of atmospheric pollutants such as ozone and secondary organic aerosol (SOA)
(Bonn et al., 2017; Heinritzi et al., 2020). Long-term BVOC emissions measurement above Sitka
spruce plantations is needed to confirm this model observation.
**Table 6 – Modelled annual isoprene, total monoterpene and total BVOC emissions per hectare of**
**SRF Sitka spruce, hybrid aspen and Italian alder plantations, using meteorology data from two**
**locations, East Grange in east Scotland, and Alice Holt in south-east England.**

| | | | Total MT / kg C ha$^{-1}$ y$^{-1}$ | Isoprene / kg C ha$^{-1}$ y$^{-1}$ | Total BVOC / kg C ha$^{-1}$ y$^{-1}$ |
|---|---|---|---|---|---|
| Sitka spruce | 2018 | East Grange | 12.3 | 18.0 | 30.3 |
| | 2019 | East Grange | 7.95 | 2.67 | 10.6 |
| | 2018 | Alice Holt | 21.2 | 30.3 | 51.5 |
| | 2019 | Alice Holt | 13.7 | 11.9 | 25.6 |
| | Average | | 13.8 | 15.7 | 29.5 |
| Hybrid aspen | 2018 | East Grange | 0.2 | 12.1 | 12.3 |
| | 2019 | East Grange | 0.3 | 13.0 | 13.3 |
| | 2018 | Alice Holt | 0.5 | 22.2 | 22.7 |
| | 2019 | Alice Holt | 0.2 | 14.8 | 15.0 |
| | Average | | 0.3 | 15.5 | 15.8 |
| Italian alder | 2018 | East Grange | 0.88 | 0.02 | 0.90 |
| | 2019 | East Grange | 0.33 | 0.01 | 0.34 |
| | 2018 | Alice Holt | 1.53 | 0.04 | 1.57 |
| | 2019 | Alice Holt | 0.52 | 0.02 | 0.54 |
| | Average | | 0.81 | 0.02 | 0.84 |




### 3.6    Uncertainties in measured and modelled fluxes
There are several uncertainties and simplifications in our approach to scaling-up from periodic
branch chamber emission measurements to annual canopy–scale predictions. We suggest that
uncertainties in the quantification of individual measurements of BVOC emissions are likely to be 16-
17% based on previous error propagation calculations (Purser et al., 2020). The nature of the
chamber measurement technique is likely to have an impact upon the BVOC emissions due to the
altered environmental conditions that may result. In addition, field-based measurements of emission
rates, collected under natural conditions for the UK but far from standard conditions (PAR 1000
$\mu$mol m$^{-2}$ s$^{-1}$, temperature 30 $^{\circ}$C) introduce an uncertainty when standardised to form emission
potentials.

Further uncertainty may then come from extrapolating these emission potentials in models for the
prediction of fluxes using measured meteorology for a given field site. The modelling undertaken
here does not include parameters such as soil moisture, humidity and wind speed as no continuous
data for these parameters were available but it is noted these would further constrain the model
estimate. In addition, there are uncertainties in collating data points to create seasonal averages for
each year, up to 25-50% based on the relative standard deviation in this case. Converting from
emissions per leaf mass to per leaf area also adds uncertainty since leaf mass:area data is highly
variable and dependent upon the tree species and sample location. However, we collected LMA data
from a range of studies in areas close to the UK with a similar climate (Table 1), and the LMA
uncertainty associated ranges from 16% to 24% RSD dependent upon tree species. The emissions
predicted from the canopy are also lacking the influence of processes such as BVOC uptake by the
forest floor, deposition to leaf surfaces and the influence of reactions with other atmospheric
chemical species such as hydroxyl, ozone and nitrogen oxides.



Emissions in early spring measured in the chambers from flowers (catkins) were not included in this
scale up exercise since only emission rates from foliage were used in the model. It is noted that
these floral emissions may contribute significantly to spring time BVOC emissions across a two or
three week time period (Baghi et al., 2012), but become less significant relative to the yearly
contribution. It should be noted that BVOC emissions are predicted by the model in winter for Sitka
spruce which maintains its canopy all year. However, this may be an over prediction of the emissions
as, on some occasions, demonstrated by our chamber measurements, winter BVOC emission may be
very low or absent from this species. Similarly, rain events have been shown to alter BVOC emissions
and may have different effects on the short term (increasing) and the longer term (decreasing),
which are also not accounted for in the model (Holzinger et al., 2006). These factors are likely to lead
to an over estimation of emissions from all species but in particular Sitka spruce on a per annum
basis.

Finally, algorithms used to scale up branch chamber emissions to canopy-level emissions have also
been suggested to give variable results, with MEGAN 2.1 typically producing lower (but perhaps
more realistic) flux estimates (Langford et al., 2017). This is an important consideration when
comparing annual estimates to total UK BVOC emissions in section 3.7 where older, more simplified
algorithms may have been applied.

### 3.7 Assessing potential impact of SRF plantation expansion on UK BVOC emissions
The annual average BVOC emissions data from section 3.5.3 (Table 6) was used to explore the
possible impact on total UK BVOC emissions arising from increased SRF planting under a suggested
bioenergy expansion in the UK (see introduction). The following estimates assume all bioenergy
expansion is SRF. However it is more likely that a combination of SRC, SRF and miscanthus could be
used in the UK for biomass and as such these estimates should be treated as a single extreme case





scenario. Meteorological data from AH and EG was used for model simulations as stated in section
3.5.2. Isoprene and monoterpene emissions are reported separately in Table 7 but also combined to
give a "total BVOC" emission.

**Table 7 – Modelled average annual emissions from 0.7 Mha of SRF expansion.**

| 0.7 Mha SRF expansion scenario | Total monoterpene / kt y$^{-1}$ | Isoprene / kt y$^{-1}$ | Total BVOC / kt y$^{-1}$ |
|---|---|---|---|
| Sitka | 9.7 | 11 | 20.7 |
| Aspen | 0.2 | 10.9 | 11.1 |
| Alder | 0.6 | 0 | 0.6 |


In the scenario of an expansion of 0.7 Mha of SRF, the total BVOC emissions from Sitka spruce SRF
could equate to 20.7 kt y$^{-1}$. For Aspen it could potentially be 11.1 kt y$^{-1}$, whilst for Italian alder it is
much smaller at 0.6 kt y$^{-1}$. These potential increases in BVOC emissions are compared in Table 8 to
current predicted annual emissions of BVOCs from vegetation in the UK. Several air quality models
have been used to estimate the total isoprene and total monoterpene emissions from UK vegetation
(AQEG, 2020), with an earlier model (Simpson et al., 1999) determining isoprene to be the dominant
BVOC emission whilst later models suggest monoterpenes dominate (Hayman et al., 2017, 2010;
Stewart et al., 2003). The meteorological data used in some of these models are limited to a single
year, e.g. 1998, where the uncertainty in the model estimates could range by a factor of 4 (Stewart
et al., 2003), whilst others are the average emissions across many years and so report a range
(Hayman et al., 2017). In addition, models of UK BVOC emissions are particularly reliant upon the
emission potential attributed to Sitka spruce as this accounts for nearly 21% of UK forest cover and,
as discussed in section 3.3.3, only a limited number of studies have been conducted on Sitka spruce
BVOC emissions. This simple impact assessment used a limited set of meteorological data to
represent two contrasting years (one warmer drier year and one cooler wetter year, relative to the
30 year average) and for two 'ends' of the British climate range of temperature and PAR: north (East
Grange, Scotland) and south (Alice Holt, England).




However, given these uncertainties, simulations of the impact of potential future land–use changes
on atmospheric BVOC emissions are important first steps to gain a better understanding of any
potential future impacts on air quality.

It is worth noting that currently the UK has an estimated 3.2 Mha of woodland, of which 0.67 Mha is
covered by Sitka spruce (Forest Research, 2020) (similar in size to the future planting scenario used
here), a small area of  alder (0.053 Mha, Forest Research, 2012) and even smaller area of aspen.
Comparing the total BVOC emissions for a 0.7 Mha SRF expansion scenario to the annual total BVOC
emissions for the UK suggests that the Sitka spruce and hybrid aspen scenarios could potentially
increase the total BVOC emissions in the ranges of 12–35% and 7–19% respectively, dependent upon
the original BVOC emission model used for this comparison (Table 8). For Italian alder this increase in
total BVOC is an order of magnitude smaller, ranging from 0.3–1%. It can therefore be suggested
that future hybrid aspen SRF plantations for bioenergy will likely emit no more BVOC than equivalent
expansion of young Sitka spruce plantations. Expansion of SRF with Italian alder may bring about no
significant changes to the UK BVOC emissions at the national level.

Any future distribution of bioenergy crops including SRF in the UK will depend on several factors
including available land, locations that are most suitable to obtain high biomass yields, locations that
are close to energy-generation plants and locations close to opportunities for $CO_2$ storage, in the
case of using BECCS to reach net-zero targets (Donnison et al., 2020).  Further work is needed to
better understand how these changes in BVOC emissions may impact air chemistry and potentially
air quality (in particular ozone and SOA) at local to UK national scale.





**Table 8 – Potential increase in isoprene, total monoterpene and total BVOC emissions from an**
**additional 0.7 Mha of SRF plantations compared to previous modelled estimates of total UK BVOC**
**emissions.**

| | Modelled UK total emissions / kt y⁻¹ | | | Sitka spruce SRF % of modelled UK emissions | | | Hybrid aspen SRF % of modelled UK emissions | | | Italian alder SRF % of modelled UK emissions | | |
|---|---|---|---|---|---|---|---|---|---|---|---|---|
| *Model Reference* | MT | Isoprene | Total | MT | Isoprene | Total | MT | Isoprene | Total | MT | Isoprene | Total |
| Simpson et al. 1999 | 30 | 58 | 88 | 32 | 19 | 24 | 0.7 | 19 | 13 | 1.9 | 0.0 | 0.7 |
| Stewart et al. 2000 | 83 | 8 | 91 | 12 | 138 | 23 | 0.3 | 136 | 12 | 0.7 | 0.2 | 0.6 |
| Hayman et al. 2010 (forest only) | 52 | 7 | 59 | 19 | 157 | 35 | 0.4 | 155 | 19 | 1.1 | 0.2 | 1.0 |
| Hayman et al. 2017 (minimum) | 110 | 33 | 143 | 9 | 33 | 14 | 0.2 | 33 | 8 | 0.5 | 0.0 | 0.4 |
| Hayman et al. 2017 (maximum) | 125 | 44 | 169 | 8 | 25 | 12 | 0.2 | 25 | 7 | 0.5 | 0.0 | 0.3 |


Values that are shown as 0.0 are < 0.05%; Hayman et al 2017 (minimum) and (maximum) values are
the upper and lower estimates of BVOC emissions published that account for yearly changes in
meteorology in the model scenarios.         Conclusions
Winter and spring emissions of isoprene and monoterpenes in the three potential short-rotation
forestry (SRF) species of Sitka spruce, hybrid aspen and Italian alder were one or two orders of
magnitude smaller than their respective emissions in summer. There were large differences in the
BVOC emission rates and compounds between the three species, with d-limonene, α-pinene and β-
myrcene being the major monoterpenes across all three species.
Sitka spruce emitted more isoprene and monoterpenes during the warmer, drier 2018 than in the
cooler, wetter 2019. Isoprene emissions for hybrid aspen were similar in both years but
monoterpene emissions were higher in 2018 compared to 2019. Italian alder did not often emit
detectable amounts of isoprene in either year, and only a little monoterpene in 2018. The observed
differences in emissions of the relative amounts of isoprene compared to monoterpenes in the case
of Sitka spruce could lead to differences in SOA generation in warmer and cooler years.
Overall, forest floor emissions of monoterpenes were a factor 10 to 1000 times smaller than the
canopy emissions. The forest floor emissions were more variable and acted as a source for most of





the time with occasional instances (<4 measurement occasions out of 20) when the forest floor
acted as a sink for monoterpenes. Further work is necessary under controlled conditions to fully
understand the drivers and components of forest floor emissions.
Total annual emissions per unit ground area for each SRF species were derived using MEGAN 2.1 and
scaled up to a 0.7 Mha future SRF expansion scenario for the UK. Under this scenario, total modelled
UK BVOC emissions (the sum of isoprene and total monoterpene emissions) could increase by <1–
35% depending on the species planted and the UK BVOC emissions model used. Future work to
understand how any increase in forest cover and BVOC emissions may impact the atmospheric
chemistry in NOx dominated regions is needed so that air quality impacts from pollutants such as
ozone can be determined across the UK.
*Author contributions.* JILM, JD and MRH conceptualized the study, acquired funds for the study,
supervised the study, and edited and reviewed the original draft. JILM gave permission for the use of
the field site at East Grange. JD provided laboratory equipment. GP contributed to the
conceptualization of the study, developed the methodology, collected field samples, conducted
measurements and analysis and wrote the original draft. RASS assisted in collection of field samples,
conducted measurements and analysis related to leaf area index at East Grange. LKD assisted with
collection of field samples and analysis.

*Competing interests.* The authors declare that they have no conflict of interest.

*Acknowledgements.* Gemma Purser acknowledges CASE doctoral training partnership funding from
the Natural Environment Research Council through grant number NE/L002558/1. The Forestry
Commission contributed to the CASE award through the climate change research programmes of
Forest Research. We would like to thank Adam Ash and Colin McEvoy of Forest Research for
assistance with the meteorological station at East Grange. From UK CEH we thank Peter Levy and
Nick Cowan for assistance with data loggers and Ben Langford for the use of the Pocket MEGAN 2.1
excel beta 3 calculator.

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
