# Peer review of "Isoprene and monoterpene emissions from alder, aspen and spruce short rotation forest plantations in the UK 2 3 Gemma Purser\*1,2, Julia Drewer1, Mathew R. Heal2, Robert A. S. Sircus2, Lara K. 4 Dunn2, James I. L. Morison3 5 6 7 <su"

_Biogeosciences, 2020_

## Referee Comment (RC1) · Andrew Leslie (Referee) · 28 Dec 2020

**General comments:**

Interesting study that is nicely written with excellent refencing – just one minor mistake. The research contributes to an area that has had relatively little work undertaken in the UK. Data collection methods are sensible.

**Individual issues:**

Figure 1,2 and 3 show error bars and yet there has been no discussion about whether the data were normally distributed. This is also the case in the text as standard deviations and errors are mentioned. If the data are non parametric then medians and interquartile ranges would be more appropriate.

The abbreviation MT is used for monoterpenes but not explained in the text.

Table 4, Table 7 Figure 1 and Figure 3 the title states that averages are presented. An average could be the mean, mode or median so use the term, 'mean'. See comment above about normality of the data.

**Specific comments:**

| Line | Comment |
|------|---------|
| 16 | This expansion rather than that expansion |
| 52 | Expand briefly on why more domestic bioenergy is required. Why cannot we import it all? |
| 54 | Willow (*Salix* spp.) |
| 69 | Give scientific names in brackets after first mention of common names of the trees. |
| 101 | growth in SRF not growth during |
| 135 | Guenther et al 2012 |
| 153 | Soil not soils |
| 193 | Three similar branches. How were these sampled to ensure they were representative? |
| 204 | Ensure a sufficient |
| 261 | Can be standardised to… is this the convention or is it a standard you have decided on? |
| 280 | replace Guenther et al. 1993 with G93 as previously stated |
| 347 | Scaling for spring and summer – do you have any justification for using 25% and 50%? |
| 356 | Justify assumption that emissions driven by temperature. Are there studies to support this? |
| 435 | Alder sp. Ie one species or alder spp. more than one species. Alder should start with a lower case a ie alder |
| 518 | Delete the Sitka spruce title in the graph |
| 544 | They may be vegetatively propagated but they have been from different crosses so there is genetic variation between individuals. Indeed there may be more genetic variation then form seed collected from an individual stand. |
| 610 | Underneath aspen (*Populus tremula*) trees…. |
| 612 | American aspen (*Populus tremuloides*) |
| 613 | Although are not chemically… |
| 670 | conifer spp. |
| 724-728 | Perhaps provide some commentary on the large disparity between your estimate of isoprene emissions and Beverland's |
| 753 | You use the abbreviations EG and AH for the two sites for the first time towards the end of the paper. Either use these consistently throughout the manuscript or not. |
| 798 | Different effects in the short term |

| 869 | Make it clearer Conclusions is a heading |

---

## Referee Comment (RC2) · Anonymous Referee #2 · 4 Jan 2021

General comments

The paper is aimed at describing how different planted trees affect air quality in Great Britain. VOC emission rates were measured in two years 2018 and 2019. The amount of samples taken was limited considering quite large variability of BVOC emissions. The goal is not reached, but this is a good start to evaluate air quality impacts of planted forests, which is an important topic now when forests are being planted for carbon sequestration purposes.

The paper is well written, uncertainties of the measurements are evaluated, earlier literature is well cited and the overall presentation is well structured and clear. The

paper is suitable for publication in Biogeosciences after minor revisons.

Specific comments

The paper is aimed at evaluating VOC emissions impact on air quality, i.e. ozone and aerosol formation, but the measurements include only isoprene, monoterpenes and oxygenated monoterpenes. Sesquiterpenes (SQT) could have been measured at the same time and their secondary organic aerosol (SOA) formation potential is much greater than that of monoterpenes. It is really pity that the SQTs are excluded from the study, they would certainly have had an impact and SQT emission rate data is overall very sparse. In addition to air quality impacts, VOC emissions have also climate impacts, other than C sequestration. SOA formed from the reactions of the VOCs impact the climate by scattering and absorbing radiation. This is beyond the scope of the current manuscript but highlights the importance of knowing also SQT emission rates.

Measurements: -It is very good that the collars were placed already previous year. This certainly reduced emissions from cut roots etc. -Usually Teflon films are used as chamber materials in VOC emission measurements. Why did you choose acrylic chambers? Did you test the suitability of acrylic chambers before the measurements that VOCs are not retained on the surfaces or for memory effects? -All VOC emissions have pronounced diurnal variation with maximum emission during the afternoon and minimum at night, mostly driven by temperature and light. Therefore, it is important to say if you use measured emission rates or standardized emission potentials. Throughout the text, please be accurate what you mean. For example, in Figure 1 and 3 captions it says emissions, but are they measured rates or standardized potentials? -I agree with the authors that measurements on canopy scale would be very useful, but the measurements of the larger VOCs would be even more important.

---

## Author Comment (AC1) · 29 Jan 2021

**bg-2020-437: Isoprene and monoterpene emissions from alder, aspen and spruce short rotation forest plantations in the UK**

**Response to reviewer Andrew Leslie**

General comments:
Interesting study that is nicely written with excellent referencing – just one minor mistake. The research contributes to an area that has had relatively little work undertaken in the UK. Data collection methods are sensible.

Response: We thank Andrew Leslie for his time spent reading our manuscript and for his comments and suggestions to improve the manuscript. Below we respond to each review comment individually (in blue font), indicating changes made to the revised manuscript.

Individual issues:

Figure 1,2 and 3 show error bars and yet there has been no discussion about whether the data were normally distributed. This is also the case in the text as standard deviations and errors are mentioned. If the data are non parametric then medians and interquartile ranges would be more appropriate.

Response: We agree the data are highly likely to be non-normal, so to provide the reader with all information we now supply the additional data of median and interquartile range in a separate table in the Supplementary Information. We retain presentation and use of mean emissions in the main manuscript as this is what is typically used in the literature (for our literature comparison) and because it is the mean that forms the basis of scale-up to area-based and annual emissions in sections 3.5-3.7. The standard deviation is still a useful summary of the degree of variability in the measurements presented. The following text was added on line 446-447.

"The equivalent median and interquartile ranges for the data collected during this study can be found in the Supplementary Information S4".

The abbreviation MT is used for monoterpenes but not explained in the text.

Response: We have now added "MT = Monoterpene" to the captions of Table 4, Table 5, Table 6 and Table 8.

Table 4, Table 7 Figure 1 and Figure 3 the title states that averages are presented. An average could be the mean, mode or median so use the term, 'mean'. See comment above about normality of the data.

Response: All instances of the word "average" in the text and relevant figure and table captions have now been replaced with the term "mean".

Specific comments:
Line Comment
16 This expansion rather than that expansion

Response: Amended as suggested.

52 Expand briefly on why more domestic bioenergy is required. Why cannot we import it all?

Response There is likely to be a significantly higher carbon cost associated with the import of biomass than with a domestic source of biomass (Ricardo Energy & Environment, 2020). If the UK (and indeed the world) is to reduce GHG emissions to the point of net zero then these additional carbon costs should be minimized. For clarification the following text was added to lines 51-54:

"However, importing biomass contributes higher carbon emissions than biomass grown in the UK (Ricardo Energy & Environment, 2020) so a larger contribution from domestic supply of bioenergy in the UK is required if the UK is to achieve net-zero."

54 Willow (Salix spp.)

Response: Amended as suggested in line 56

69 Give scientific names in brackets after first mention of common names of the trees.

Response: The text has now been replaced with the following text in lines 71-76: "(hybrid aspen (*Populus tremula L. x tremuloides* Michx.), red alder (*Alnus rubra* Bong.), common alder (*Alnus glutinosa* (L.) Gaertn), Italian alder (*Alnus cordata* Desf.), sycamore (*Acer pseudoplatanus*), Sweet chestnut (*Castanea sativa* Mill.), eucalyptus spp. (*Eucalyptus gunnii, Eucalyptus nitens (Vic), Eucalyptus nitens (NSW), E. glaucescens*) and the two conifer species Sitka spruce (*Picea sitchensis* Bong. Carr) and hybrid larch (*Larix x marschlinsii Coaz*) (Harrison, 2010)"

101 growth in SRF not growth during

Response: Amended as suggested.

135 Guenther et al 2012

Response: Amended as suggested.

153 Soil not soils

Response: Amended as suggested.

193 Three similar branches. How were these sampled to ensure they were representative?

Response: A line has been added to the text in lines 201-203 for further clarification:

"The branches were selected to be of similar size and in a similar position on the tree. All branches were approximately 1.5 m from the ground and in a south-facing position."

204 Ensure a sufficient

Response: Amended as suggested

261 Can be standardised to… is this the convention or is it a standard you have decided on?

Response: The emission rates were standardised to 30 $^{\circ}$C and 1000 mol m$^{-2}$ s$^{-1}$ based on the findings of Guenther et al (1993, 1991). This standardisation is almost universally used by the biosphere-atmosphere community to enable the comparison of BVOC emissions from different species measured under different ambient conditions. Clarification is now made by replacing "can be" to "were" in line 271.

280 replace Guenther et al. 1993 with G93 as previously stated

Response: Amended as suggested in line 292.

347 Scaling for spring and summer – do you have any justification for using 25% and 50%?

Response: In the absence of multiple LAI measurements taken across the year at the East Grange site the observations of a deciduous forest in the UK by Ogunbadewa et al.(2012) were adopted as the basis for this scaling used in the modelling part of our study. To make this clear to the reader the manuscript has been modified on lines 327-332 to:

"In measurements of LAI by Ogunbadewa et al. (2012), taken across a year in a deciduous forest in the UK, the LAI was at its maximum by July and during spring the LAI increased such that it was around a quarter of the maximum by late April and around a half by mid-May. These seasonal changes in LAI were therefore adopted for use in the MEGAN 2.1 model (Table 2) in the absence of multiple seasonal LAI measurements taken at East Grange during our study."

356 Justify assumption that emissions driven by temperature. Are there studies to support this?

Response: Additional references have been added and the text reworded for clarification in lines 401-406 as follows:

"Although some individual monoterpene compounds may be produced in the leaves in response to light and temperature to varying degrees, due to the use of the collective "total monoterpenes" as a model input the simplification was used that monoterpene emissions were driven by temperature only and no light specific emission factor was applied (Guenther et al., 2006, 1993)."

435 Alder sp. Ie one species or alder spp. more than one species. Alder should start with a lower case a ie alder

Response: "alder spp." has been added to line 456-457 in manuscript.

518 Delete the Sitka spruce title in the graph

Response: Amended in manuscript.

544 They may be vegetatively propagated but they have been from different crosses so there is genetic variation between individuals. Indeed there may be more genetic variation then form seed collected from an individual stand.

Response: The text has been modified to make our suggestion more speculative, lines 566-567:

"This may reflect that the trees grown via vegetative propagation could be from a genetically similar source."

610 Underneath aspen (Populus tremula) trees….

Response: Amended in manuscript in line 633.

612 American aspen (Populus tremuloides)

Response: Amended in manuscript in line 635.

613 Although are not chemically…

Response: Amended in manuscript in line 636.

670 conifer spp.

Response: Amended in manuscript in line 689.

724-728 Perhaps provide some commentary on the large disparity between your estimate of isoprene emissions and Beverland's

Response: The following additional clarification as to why there may have been a large disparity is added to lines 252-254.

"These emissions are much lower than our model estimates although it was reported that there were analytical difficulties with the micrometeorological techniques and limited data which could account for this disparity."

753 You use the abbreviations EG and AH for the two sites for the first time towards the end of the paper. Either use these consistently throughout the manuscript or not.

Response: The instance where EG and AH have been used are now replaced with "East Grange" and "Alice Holt" respectively for consistency with the earlier part of the manuscript. The amendments apply to lines:770, 771, 780, 782-784

798 Different effects in the short term

Response: Amended in manuscript on line 837.

869 Make it clearer Conclusions is a heading

Response: The conclusion title has now been repositioned to line 909 and given the section heading number 4.

**References**

Guenther, A., Karl, T., Harley, P., Wiedinmyer, C., Palmer, P.I., Geron, C., 2006. Estimates of global terrestrial isoprene emissions using MEGAN (Model of Emissions of Gases and Aerosols from Nature). Atmos. Chem. Phys. 6, 3181–3210. https://doi.org/10.5194/acp-6-3181-2006

Guenther, A.B., Monson, R.K., Fall, R., 1991. Isoprene and monoterpene emission rate variability: Observations with eucalyptus and emission rate algorithm development. J. Geophys. Res. Atmos. 96, 10799–10808. https://doi.org/https://doi.org/10.1029/91JD00960

Guenther, A.B., Zimmerman, P.R., Harley, P.C., Monson, R.K., Fall, R., 1993. Isoprene and monoterpene emission rate variability: model evaluations and sensitivity analyses. J. Geophys. Res. 98. https://doi.org/10.1029/93jd00527

Ogunbadewa, E.Y., 2012. Tracking seasonal changes in vegetation phenology with a SunScan canopy analyzer in northwestern England. Forest Sci. Technol. 8, 161–172. https://doi.org/10.1080/21580103.2012.704971

Ricardo Energy & Environment, 2020. Analysing the potential of bioenergy with carbon capture in the UK to 2050 [WWW Document]. URL https://www.gov.uk/government/publications/the-potential-of-bioenergy-with-carbon-capture

---

## Author Comment (AC2) · 29 Jan 2021

**bg-2020-437: Isoprene and monoterpene emissions from alder, aspen and spruce short rotation forest plantations in the UK**

**Response to Anonymous Referee #2**

General comments
The paper is aimed at describing how different planted trees affect air quality in Great Britain. VOC emission rates were measured in two years 2018 and 2019. The amount of samples taken was limited considering quite large variability of BVOC emissions. The goal is not reached, but this is a good start to evaluate air quality impacts of planted forests, which is an important topic now when forests are being planted for carbon sequestration purposes.
The paper is well written, uncertainties of the measurements are evaluated, earlier literature is well cited and the overall presentation is well structured and clear. The C1 paper is suitable for publication in Biogeosciences after minor revisons.

Response: We thank Anonymous Referee #2 for the time spent reading our manuscript and the positive comments and suggestion we received to improve it. Below we respond to each review comment individually (in blue font), indicating changes made to the revised manuscript.

Specific comments
The paper is aimed at evaluating VOC emissions impact on air quality, i.e. ozone and aerosol formation, but the measurements include only isoprene, monoterpenes and oxygenated monoterpenes. Sesquiterpenes (SQT) could have been measured at the same time and their secondary organic aerosol (SOA) formation potential is much greater than that of monoterpenes. It is really pity that the SQTs are excluded from the study, they would certainly have had an impact and SQT emission rate data is overall very sparse. In addition to air quality impacts, VOC emissions have also climate impacts, other than C sequestration. SOA formed from the reactions of the VOCs impact the climate by scattering and absorbing radiation. This is beyond the scope of the current manuscript but highlights the importance of knowing also SQT emission rates.

Response: We thank the reviewer for highlighting these important points and we appreciate the impact that sesquiterpenes may have on air quality. We did not actively exclude sesquiterpenes from this study, it was just not possible to include them in this first assessment. Given this is just a first look assessment of the impacts of BVOC from bioenergy forests it is our hope that this work will be extended in the future to include the much needed sesquiterpene data that is significantly lacking in the databases of BVOC emission potentials for a range of short-rotation forest species relevant to UK bioenergy.

Measurements: -It is very good that the collars were placed already previous year. This certainly reduced emissions from cut roots etc.

Response: We appreciate the positive comment with respect to this aspect of the methodology.

-Usually Teflon films are used as chamber materials in VOC emission measurements. Why did you choose acrylic chambers? Did you test the suitability of acrylic chambers before the measurements that VOCs are not retained on the surfaces or for memory effects?

Response: Although we did not specifically test the surface effects of VOCs on the chamber materials we note that polymethyl methacrylate, commonly known as acrylic, plexiglass and Perspex glass has previously been used for the construction of chambers in both a full (Ghirardo et al., 2012; Potosnak et al., 2013; Spielmann et al., 2017) or partial capacity (Ghirardo et al., 2011) for BVOC emission measurements. In some cases the acrylic chamber has been coated in an inert Teflon film to prevent the losses of BVOC to reduce the absorption and adsorption of BVOCs to the chamber walls (Aalto et al., 2015). We appreciate acrylic may be less inert than Teflon and therefore BVOC emissions could be subject to interference from adsorption/desorption processes. We used a dynamic system in our study and equilibrated with flow through for up to 30 minutes before sampling to reduce the potential effects of chamber material interferences as shown in a previous study by Stewart-Jones & Poppy (2006).

All VOC emissions have pronounced diurnal variation with maximum emission during the afternoon and minimum at night, mostly driven by temperature and light. Therefore, it is important to say if you use measured emission rates or standardized emission potentials.
Throughout the text, please be accurate what you mean. For example, in Figure 1 and 3 captions it says emissions, but are they measured rates or standardized potentials?

Response: We have now clarified each instance of the term emissions throughout the manuscript as to whether it refers to measured or standardized emissions.

I agree with the authors that measurements on canopy scale would be very useful, but the measurements of the larger VOCs would be even more important.

Response: We thank the reviewer for highlighting this important point. We believe both these points are important and the following additional statement has been added to emphasise this in lines 599-603.

"….Norway spruce has also been found to be significant emitters of sesquiterpenes (Hakola et al., 2017). Given the dominance of Sitka spruce plantations in the UK (and Ireland), the potential for variation within this species, and the limited literature data on BVOC emissions, we suggest further measurements are needed at the branch and canopy level to fully assess the terpenoid species composition and their subsequent impact on air quality."

References

Aalto, J., Porcar-Castell, A., Atherton, J., Kolari, P., Pohja, T., Hari, P., Nikinmaa, E., Petäjä, T., Bäck, J., 2015. Onset of photosynthesis in spring speeds up monoterpene synthesis and leads to emission bursts. Plant Cell Environ. 38, 2299–2312. https://doi.org/10.1111/pce.12550

Ghirardo, A., Gutknecht, J., Zimmer, I., Brüggemann, N., Schnitzler, J.P., 2011. Biogenic volatile organic compound and respiratory $CO_2$ emissions after 13C-labeling: Online tracing of C translocation dynamics in poplar plants. PLoS One 6, 2–5. https://doi.org/10.1371/journal.pone.0017393

Ghirardo, A., Heller, W., Fladung, M., Schnitzler, J.P., Schroeder, H., 2012. Function of defensive volatiles in pedunculate oak (Quercus robur) is tricked by the moth Tortrix viridana. Plant, Cell Environ. 35, 2192–2207. https://doi.org/10.1111/j.1365-3040.2012.02545.x

Hakola, H., Tarvainen, V., Praplan, A.P., Jaars, K., Hemmilä, M., Kulmala, M., Bäck, J., Hellén, H., 2017. Terpenoid and carbonyl emissions from Norway spruce in Finland during the growing season. Atmos. Chem. Phys. 17, 3357–3370. https://doi.org/10.5194/acp-17-3357-2017

Potosnak, M.J., Baker, B.M., Lestourgeon, L., Disher, S.M., Griffin, K.L., Bret-Harte, M.S., Starr, G., 2013. Isoprene emissions from a tundra ecosystem. Biogeosciences 10, 871–889. https://doi.org/10.5194/bg-10-871-2013

Spielmann, F.M., Langebner, S., Ghirardo, A., Hansel, A., Schnitzler, J.P., Wohlfahrt, G., 2017. Isoprene and α-pinene deposition to grassland mesocosms. Plant Soil 410, 313–322. https://doi.org/10.1007/s11104-016-3009-8

Stewart-Jones, A., Poppy, G.M., 2006. Comparison of glass vessels and plastic bags for enclosing living plant parts for headspace analysis. J. Chem. Ecol. 32, 845–864. https://doi.org/10.1007/s10886-006-9039-6